# Time varying parameter models for catchments with land use change: the importance of model structure

Sahani Pathiraja[1,2], Daniela Anghileri[3], Paolo Burlando[3],

Ashish Sharma[2] Lucy Marshall[2], Hamid Moradkhani[4]

[1]Institut für Mathematik

Universität Potsdam

Potsdam

GERMANY

Email: pathiraja@uni-potsdam.de

[2]Water Research Centre

School of Civil and Environmental Engineering

University of New South Wales

Sydney, NSW

AUSTRALIA

[3]Institute of Environmental Engineering

ETH Zurich

Zurich

SWITZERLAND

[4]Department of Civil, Construction and Environmental Engineering

University of Alabama

Tuscaloosa, Alabama

USA

**Abstract**
Rapid population and economic growth in South-East-Asia has been accompanied by extensive land
use change with consequent impacts on catchment hydrology. Modelling methodologies capable of
handling changing land use conditions are therefore becoming ever more important, and are
receiving increasing attention from hydrologists. A recently developed Data Assimilation based
framework that allows model parameters to vary through time in response to signals of change in
observations is considered for a medium sized catchment (2880 km$^2$) in Northern Vietnam
experiencing substantial but gradual land cover change. We investigate the efficacy of the method
as well as the importance of the chosen model structure in ensuring the success of a time varying
parameter method. The method was used with two lumped daily conceptual models (HBV and
HyMOD) that gave good quality streamflow predictions during pre-change conditions. Although both
time varying parameter models gave improved streamflow predictions under changed conditions
compared to the time invariant parameter model, persistent biases for low flows were apparent in
the HyMOD case. It was found that HyMOD was not suited to representing the modified baseflow
conditions, resulting in extreme and unrealistic time varying parameter estimates. This work shows
that the chosen model can be critical for ensuring the time varying parameter framework
successfully models streamflow under changing land cover conditions. It can also be used to
determine whether land cover changes (and not just meteorological factors) contribute to the
observed hydrologic changes in retrospective studies where the lack of a paired control catchment
precludes such an assessment.

# 1. Introduction

Population and economic growth in South-East Asia has led to significant land use change, with rapid deforestation occurring largely for agricultural purposes [*Kummer and Turner*, 1994]. Forest cover in the Greater Mekong Sub-region (comprising Myanmar, Thailand, Cambodia, Laos, Vietnam, and South China) has decreased from about 73% in 1973 to about 51% in 2009 [*WWF*, 2013]. Vietnam in particular has had the second highest rate of deforestation of primary forest in the world, based on estimates from the Forest Resource Assessment by the United Nations Food and Agriculture Organization [*FAO*, 2005]. Such extensive land use change has the potential to significantly alter catchment hydrology (in terms of both quantity and quality), with its effects sometimes not immediate but occurring gradually over a lengthy period of time. Recent estimates from satellite measurements indicate that rapid deforestation continues in the region, although at lower rates [e.g. *Kim et al.,* 2015]. Persistent land use change necessitates modelling methodologies that are capable of providing accurate hydrologic forecasts and predictions, despite non-stationarity in catchment processes. This is also particularly relevant for water resource management which requires reliable estimates of water availability, both in terms of volume and timing, to properly allocate the resource between different water uses and to prevent flood damages. Vietnam has built many reservoirs in the last decades and more are planned because they are considered to be fundamentally important for electricity production, flood control, water supply and irrigation, ultimately contributing to the development of the country [*Giuliani et al*., 2016].

The literature on land-use change and its impacts on catchment hydrology is extensive, with studies examining the effects of 1) conversion to agricultural land-use [*Thanapakpawin et al*, 2007; *Warburton et al.,* 2012]; 2) deforestation [*Costa et al.,* 2003; *Coe et al*, 2011]; 3) afforestation [e.g. *Yang et al., 2012*; *Brown et al*, 2013] and 4) urbanization [*Bhaduri et al.,* 2001; *Rose & Peters*, 2001]. Fewer studies have examined how traditional modelling approaches must be modified to handle

non-stationary conditions, or how modelling methods can be used to assess impacts of land use
change.  Split sample calibration has been used frequently to retrospectively examine changes to
model parameters due to land use or climatic change [*Seibert & McDonnell, 2010; Coron et al., 2012;*
*McIntyre & Marshall*, 2010; *Legesse et al*, 2003].  Several other studies have employed scenario
modelling, whereby hydrologic models are parameterized to represent different possible future land
use conditions [e.g. *Niu & Sivakumar*, 2013; *Elfert & Borman*, 2010].  A related approach involves
combining land use change forecast models with hydrologic models [e.g. *Wijesekara et al.,* 2012].
However, the aforementioned approaches are unsuited to hydrologic forecasting in changing
catchments, as the predicted land use change may not reflect actual changes.  A potentially more
suitable approach in such a setting is to allow model parameters to vary in time, rather than
assuming a constant optimal value or stationary probability distribution. Many existing methods
utilising such a framework require some *apriori* knowledge of the land use change in order to inform
variations in model parameters (see for instance *Efstratiadis,* 2015; *Brown et al.,* 2006; and *Westra et*
*al.,* 2014).  Recent efforts have examined the potential for time varying parameter models to
automatically adapt to changing conditions using information contained in hydrologic observations
and sequential Data Assimilation, without requiring explicit knowledge of the changes [see for
example *Taver et al.,* 2015, *Pathiraja et al.,* 2016a&b].  Such approaches can objectively modify
model parameters in response to signals of change in observations in real time, whilst simultaneously
providing uncertainty estimates of parameters and streamflow predictions.  They can also be used to
determine whether land cover changes (and not solely meteorological factors) contribute to
observed changes in streamflow dynamics in retrospective studies where the lack of a paired control
catchment precludes such an assessment.

*Pathiraja et al.* [2016a] presented an Ensemble Kalman Filter based algorithm (the so-called Locally
Linear Dual EnKF) to estimate time variations in model parameters.  The method sequentially
assimilates observations into a numerical model in real time to generate improved estimates of

model states, fluxes and parameters based on their respective uncertainties. Its purpose is to infer

changes to catchment properties (e.g. land cover change) from hydrologic observations, without

prior knowledge of such changes, at the time scale of the available observations. It can therefore be

used for various applications: 1) to retrospectively estimate time variations in model parameters; 2)

for short-term predictive modelling (days to weeks), e.g. flood forecasting; and 3) for on-line/real

time water resource management, e.g. determining releases from reservoirs in catchments with

changing land cover conditions. In retrospective mode, the method is advantageous compared to

split-sample calibration type approaches since no *apriori* knowledge of land use change is needed,

and the modeller does not have to make somewhat arbitrary decisions about how to segregate the

data. When used for prediction or forecasting, states and parameters are updated sequentially using

all available observations up until the current time. These updated states and parameters are then

used along with the prior parameter generating model to produce hydrologic predictions over a short

time horizon. This allows one to seamlessly obtain predictions without the modeller needing to

explicitly modify the model to account for any catchment changes. The efficacy of the method was

demonstrated in *Pathiraja et al.* [2016b] through an application to small experimental catchments (<

350 ha) with drastic land cover changes and strong signals of change in streamflow observations.

Here we investigate two issues related to the use of time varying parameter models for prediction in

realistic catchments with changing land cover conditions. Firstly, we investigate the efficacy of the

time varying parameter method for sparsely observed, medium-sized catchments with spatially

complex and gradual land use change (occurring over months/years). Several authors have

demonstrated that impacts of land use change on the hydrologic response are dependent on many

factors including the type and rate of land cover conversion as well the spatial pattern of different

land uses within the catchment [*Dwarakish & Ganasri*, 2015; *Warburton et al.*, 2012]. In such

situations, the effects of unresolved spatial heterogeneities in model inputs (e.g. rainfall) and the

relatively less pronounced changes in land surface conditions make time varying parameter detection

and accurate hydrologic prediction more difficult.   The second objective is to examine the role of
the hydrologic model in determining the ability of the time varying parameter framework to provide
high quality predictions in changing conditions.  Often there may be several candidate hydrologic
models (with time invariant parameters) that have similar predictive performance for a catchment
when calibrated and validated over a time series of static land cover conditions [*Marshall et al.,*
*2006*].  This work examines whether all such candidate models in time varying parameter mode are
also capable of providing accurate predictions under changing conditions.

These issues are investigated for the Nammuc catchment (2880 km$^2$) in Northern Vietnam which has
experienced deforestation largely due to increasing agricultural development.  It serves as an ideal
test catchment to study the efficacy of the time varying parameter algorithm due to its size, spatially
complex pattern of land use changes, and lack of information on the precise timing of such changes.
Land cover change is estimated to have occurred at varying rates, with cropland accounting for
roughly 23% between 1981 and 1994, and 52% by 2000.  We also consider two lumped conceptual
hydrologic models (given the availability of point rainfall, temperature, and streamflow data)
operating at daily time step to address the second objective.  Both models demonstrate similar
performance in representing streamflow at the outlet during the pre-change calibration period
(1975-1979), although their performance during/after land use change is unknown. Therefore, the
effect of the model structure (i.e. model equations) on hydrologic predictions from the time varying
parameter models is studied.  This work represents the first application of a continuously time
varying parameter approach for modelling a real medium sized catchment with no *apriori* (or partial)
knowledge of the type and timing of land use change.

The remainder of this paper is structured as follows. Details of the study catchment and the impact
of land cover change are analysed in Section 2.  Section 3 summarizes the experimental setup
including the hydrological models and the time varying parameter estimation method used.  Results
are provided in Section 4, along with an analysis of whether the time varying model structures reflect
the observed catchment dynamics.  Finally, we conclude with a summary of the main outcomes of
the study as well as proposed future work.

## 2.  The Nammuc Catchment

The Nammuc catchment (2880 km$^2$) is located in the Red River Basin, the second largest drainage
basin in Vietnam which also drains parts of China and Laos.  The local climate is tropical monsoon
dominated with distinct wet (May to October) and dry (November to April) seasons.  The wet season
tends to have high temperatures (on average 27 to 29 °C) due to south-south easterly winds that
bring humid air masses.  Conversely, during the dry season, circulation patterns reverse carrying
cooler dry air masses to the basin (leading to average temperatures of 16 to 21°C).  Streamflow
response is consequently monsoon driven, with high flows occurring between June and October
(generally peaking in July/August) and low flows in the December to May period (Vu, 1993).  Average
annual rainfall at Nammuc varies between 1300 and 2000 mm (on average 1600 mm) and catchment
elevation ranges between 350 and 1500 m asl.  A summary of catchment properties is provided in
Table 1 for pre-change (prior to 1994) and post-change (after 1994) conditions.  This separation was
based on available land cover information as described below.

### 2.1.Data & Land Cover Change

Figure 1 shows the available land cover information for the Nammuc catchment.  Land cover
information for the catchment is scant, we were able to locate only two sources which unfortunately
do not give a complete picture over the entire time period of interest (1970 to 2004).  The first land
cover map refers to the period 1981-1994 and was obtained by the Vietnamese Forest Inventory and
Planning Institute (http://fipi.vn/Home-en.htm). The second land cover map refers to year 2000 and
was obtained from the FAO Global Land Cover database
(http://www.fao.org/geonetwork/srv/en/metadata.show?id=12749&currTab=simple). A comparison
of the two maps shows a reduction in forest cover in favor of cropland; Evergreen Leaf decreases
from about 60% to 30% whilst cropland increases from about 23% to 52%. The change in land cover
is patchy, although mostly concentrated in the northern part of the catchment. Because of the scant
information available, it is not easy to identify the precise time period of these changes. Based on the
available land cover map information and the changes to observed runoff (see Section 2.2), we posit
that a period of rapid extensive deforestation occurred in early to mid-1990s.

Daily point rainfall data is available at four precipitation stations surrounding the catchment (Dien
Bien, Tuan Giao, Quynh Nhai and Nammuc, see Figure 1).  Catchment averaged rainfall was
developed as a weighted sum of the four stations with weights determined by Thiessen Polygons.
Daily mean temperature was calculated in a similar fashion using temperature records from the 2
closest gauges (Lai Chau and Quynh Nhai, see Figure 1).  This was used to estimate Potential
Evapotranspiration through the empirical temperature-latitude based Hamon PET method [*Hamon*,
1961].  Daily rainfall, temperature and streamflow data was provided by the Vietnamese Institute of
Water Resources Planning.
## 2.2.Impact of Land Cover Change on Streamflow
The annual runoff/direct runoff coefficient and Baseflow Index were used to assess the impact of
land cover change on the hydrologic regime.  Baseflow was estimated using the two parameter
recursive baseflow filter of *Eckhardt* [2005] (see equation 1), with on-line updating of baseflow
estimates to match low flows:

$$b_k = \frac{1}{(1 - a.BFI_{max})}[(1 - BFI_{max}).a.b_{k-1} + (1 - a).BFI_{max}.y_k] \qquad (1)$$

where $b_k$ is the estimated baseflow at time $k$, $y_k$ is the total observed streamflow at time $k$, $BFI_{max}$
is the maximum value of the BFI (long term ratio of baseflow to total streamflow) and $a$ is a filter
parameter.  In this study, we adopt $BFI_{max} = 0.5$ and $a = 0.988$ based on manual optimization.

An examination of the observed streamflow and rainfall records shows that distinct changes to the
hydrologic regime are evident after the mid-1990s. The annual runoff coefficient $\left(\frac{runoff}{rainfall}\right)$ varies
between 0.4 and 0.6 prior to 1994, after which it increases to between 0.6 and 0.8 until 2004 (see
Figure 2a). However, increases to annual yields are driven mostly by changes to baseflow volume.
This is evident in Figure 2a, which shows that the increase in the annual direct runoff coefficient
$\left(\frac{runoff-baseflow}{rainfall}\right)$ is less than the increase in the total runoff coefficient (roughly 0.1 increase
compared to 0.2 respectively). A small increase in the Annual Baseflow Index $\left(\frac{baseflow}{runoff}\right)$ is apparent
also, from about 0.32 on average in the period 1970 to 1982 to 0.39 on average after 1994 (Figure
2b). This indicates that the annual increases to baseflow volume exceed the increases to direct
runoff volume. Similar changes were found by *Wang et al.* [2012] who analyzed records in the
entire Da River basin which drains the largest river in the Red River catchment. The exact physical
processes behind the observed increase in baseflow are not precisely known, particularly since
effects of land use change from forest to cropland are not unequivocal [*Price*, 2011]. Deforestation
may be associated to an increase in mean annual flow and baseflow because of lower interception
and evapotranspiration rates [e.g., *Keppeler and Ziemer*, 1990]. Nevertheless, permanent forest
removal may decrease baseflow because of soil compaction and lower infiltration rates [e.g.,
*Zimmermann et al.,* 2006; *Bormann and Klaassen;* 2008]. Some authors also show that tillage
practices, associated to forest conversion to cropland, can increase soil porosity, soil water content,
and infiltration, thus ultimately contributing to baseflow formation [e.g., *Alam et al.,* 2014].

At a seasonal time scale, it is apparent that both wet and dry season flows exhibit temporal
variations. We utilized the Moving Average Shifting Horizon (MASH) [*Anghileri et al.,* 2014] and
Mann-Kendall test to assess seasonal trends in observed streamflow, precipitation, and temperature
data. The MASH tool can be used to qualitatively assess inter-annual variations in the seasonal
pattern of a variable. It works by calculating a statistic of the data (e.g. mean) over the same block of
days in consecutive years. A steady increase in baseflow is again apparent (see February to April in
Figure 2c), as well as increases to wet season flows (see June to September in Figure 2c). Mann-
Kendall test (with significance level equal to 5%) on annual and monthly streamflow time series
shows increasing trends in almost all months, i.e., from October to July. No concurrent increases are
apparent in rainfall (see Figure 2d). Also, the Mann-Kendall test applied to precipitation time series
does not show any statistically significant trend, except a decrease in September for Nammuc and
Quynh Nhai station and an increase in July for Dien Bien station. Temperature variations are not
evident from the MASH analysis (not shown) and no significant trend can be detected by applying the
Mann-Kendall test. These results indicate that changes in streamflow dynamics are likely due to land
use change rather than climatic impacts.
## 3. Experimental Setup
### 3.1. Hydrologic Models
Conceptual lumped models operating at a daily time step were adopted due to the availability of
point rather than distributed hydro-meteorological data of sufficient length. We considered the
HyMOD [*Boyle*, 2001] and Hydrologiska Byrans Vattenbalansavdelning (HBV) [*Bergstrom et al.,* 1995]
models. They differ mainly in the way components of the response flow are separated (HBV has near
surface flow, interflow, and baseflow components whilst HyMOD has a quickflow and slow flow
component only) and how these flows are routed. A schematic of the models is shown in Figure 3.

In the HyMOD model, spatial variations in catchment soil storage capacity are represented by a
Pareto distribution with shape parameter $b$ and maximum point soil storage depth $c_{max}$. Excess
rainfall ($V$) is partitioned into three cascading tanks representing quick flow and a single slow flow
store through the splitting parameter $\alpha$. Outflow from these linear routing tanks is controlled by
parameters $k_q$ (for the quick flow stores) and $k_s$ (for the slow flow store). The model has a total of 5
states and 5 parameters.

In the HBV model, input to the soil store is represented by a power-law function (see Figure 3, note
the snow store is neglected for this study). Excess rainfall enters a shallow layer store which
generates: 1) near surface flow ($q_0$) whenever the shallow store state ($stw1$) is above a threshold
($hl1$) and 2) interflow ($q_1$) by a linear routing mechanism controlled by the $K1$ parameter.
Percolation from the shallow layer store to the deep layer store (controlled by $perc$ parameter) then
leads to the generation of baseflow also via linear routing (controlled by the $K2$ parameter). Finally, a
triangular weighting function of base length $Maxbas$ is used to route the sum of all three flow
components. There are a total of 9 parameters and 3 states.

The Shuffled Complex Evolution Algorithm (SCE-UA) [*Duan et al.*, 1993] was used to calibrate HyMOD
and the Borg Evolutionary Algorithm [*Hadka & Reed*, 2013] was used to calibrate HBV. The
calibration algorithms were selected based on previous studies that had successfully used them for
calibration of these models [*Reed et al.,* 2013; *Moradkhani et al.,* 2005]. The calibration procedure
itself is however not critical in our study, because the optimal parameter values are only used as
initial values for the time varying parameter method. Both models were calibrated to pre-change
conditions. The period 1973 to 1979 was selected for calibration (with 2 years for spin-up) as it was
expected to have minimal land cover changes (and is therefore representative of pre-change
conditions), and also to ensure sufficient data on pre-change conditions is available for assimilation.
Both models had very similar performance in terms of reproducing observed runoff (a Nash Sutcliffe
Efficiency of 0.75 and 0.77 for HyMOD and HBV respectively). HBV was slightly better at reproducing
low flows whilst HyMOD was slightly better at mid-range flows (see Table 2). Here the low flow
threshold was defined as the average annual 50[th] percentile flow and the high flow threshold as the
average annual 85[th] percentile flow.

## 3.2. Time Varying Parameter Estimation

A Data Assimilation based framework for estimating time varying parameters was presented in *Pathiraja et al.* [2016a]. The approach relies on an Ensemble Kalman Filter (EnKF) [*Evensen*,1994] to perform sequential joint state and parameter updating. EnKFs were developed to extend the applicability of the celebrated Kalman Filter [*Kalman*, 1960] to non-linear systems, although they provide a sub-optimal update as only the mean and covariance are considered in generating the posterior. However, they have been used with much success in many hydrologic applications [see for example *Reichle et al.*, 2002; *Gu et al.*, 2005; *Komma et al.*, 2008; *Sun et al.*, 2009; *Xu et al.*, 2016]. EnKFs offer a practical alternative to Sequential Monte Carlo/Particle Filter methods that propagate the full probability density through time, but suffer from several implementation issues even in moderate dimensional systems. The Locally Linear Dual EnKF method of *Pathiraja et al.* [2016a] works by sequentially proposing parameters, updating these using the Ensemble Kalman filter and available observations, and subsequently using these updated parameters to propose and update model states. An approach for proposing parameters in the time varying setting was also presented, for cases where no prior knowledge of parameter variations is available. The method was verified against multiple synthetic case studies as well as for 2 small experimental catchments experiencing controlled land use change [*Pathiraja et al.*, 2016a and *Pathiraja et al.*, 2016b]. The algorithm is summarised below, for full details refer to *Pathiraja et al.* [2016a].

### 3.2.1. Locally Linear Dual EnKF

Suppose a dynamical system can be described by a vector of states $x_t$ and outputs $y_t$ and a vector of associated model parameters $\theta_t$ at any given time *t*. The uncertain system states and parameters are represented by an ensemble of states $\{x_t^i\}_{i=1:n}$ and parameters $\{\theta_t^i\}_{i=1:n}$ each with *n* members. The prior state and parameter distributions $\{x_t^{i-}\}_{i=1:n}$ and $\{\theta_t^{i-}\}_{i=1:n}$ respectively represent our prior knowledge of the system, usually derived as the output from a numerical model. Suppose also that the system outputs are observed $(y_t^o)$ but that there is also some uncertainty associated with

these observations. The purpose of the data assimilation algorithm (here the EnKF) is to combine the
prior estimates with measurements, based on their respective uncertainties, to obtain an improved
estimate of the system states and parameters. A single cycle of the Locally Linear Dual EnKF
procedure for a given time $t$ is undertaken as follows. Note in the following, the overbar notation is
used to indicate the ensemble mean.

1. ***Propose a prior parameter ensemble.*** This involves generating a parameter ensemble using

prior knowledge. In this case, our prior knowledge comes from the updated parameter

ensemble from the previous time ($\boldsymbol{\theta}_{t-1}^{i+}$) and how it has changed over recent time steps. The

assumed parameter dynamics is a Gaussian random walk with time varying mean and

variance, given by:

$$\boldsymbol{\theta}_t^{i-} \sim N\left(\boldsymbol{\theta}_{t-1}^{i+} + \boldsymbol{m}_t.\Delta t, s^2 \boldsymbol{\Sigma}_{t-1}^{\theta}\right) \; for \; i = 1:n \tag{2}$$

$$\boldsymbol{\Sigma}_{t-1}^{\theta} = \frac{1}{n-1}\sum_{i=1}^{n}\left(\boldsymbol{\theta}_{t-1}^{i+} - \overline{\boldsymbol{\theta}_{t-1}^{+}}\right)\left(\boldsymbol{\theta}_{t-1}^{i+} - \overline{\boldsymbol{\theta}_{t-1}^{+}}\right)^{\mathrm{T}} \tag{3}$$

where $\boldsymbol{\Sigma}_{t-1}^{\theta}$ is the sample covariance matrix of the updated parameter ensemble at time $t-$

$1$; $\overline{\boldsymbol{\theta}_{t-1}^{+}}$ indicates the ensemble mean of the updated parameters at time $t-1$; $(\ )^{\mathrm{T}}$

represents the transpose operator; and $s^2$ is a tuning parameter. The prior ensemble mean

is determined as the linear extrapolation of the updated ensemble means from the previous

two time steps, i.e.:

$$\boldsymbol{m}_t[k] = \begin{cases}\boldsymbol{m}_{t-1}[k], & |\boldsymbol{m}_{t-1}[k]| \leq m_{max} \\ \boldsymbol{m}_{t-2}[k], & |\boldsymbol{m}_{t-1}[k]| > m_{max}\end{cases} \tag{4}$$

$$\boldsymbol{m}_{t-1} = \frac{\overline{\boldsymbol{\theta}_{t-1}^{+}} - \overline{\boldsymbol{\theta}_{t-2}^{+}}}{\Delta t} \tag{5}$$

$$\boldsymbol{m}_{t-2} = \frac{\overline{\boldsymbol{\theta}_{t-2}^{+}} - \overline{\boldsymbol{\theta}_{t-3}^{+}}}{\Delta t} \tag{6}$$

where $\boldsymbol{m}_t[k]$ indicates the kth component of the vector $\boldsymbol{m}_t$, the estimated rate of change.

Note that the extrapolation is forced to be less than a pre-defined maximum rate of change

$m_{max}$ to minimise overfitting and avoid parameter drift due to isolated large updates. The

maximum rate of change is model specific and will depend on the modeller's judgement
regarding expected extreme changes.
2. **Consider observation and forcing uncertainty.** This is done by perturbing measurements of
forcings and system outputs with random noise sampled from a distribution representing the
uncertainty in those measurements. The result is an ensemble of forcings ($u_t^i$) and
observations ($y_t^i$) each with $n$ members. For example, if random errors in measurements of
system outputs (herein also referred as observations) are characterized by a zero mean
Gaussian distribution, the ensemble of observations is given by:

$$y_t^i \sim N\left(y_t^o, \Sigma_t^{y^o y^o}\right) \ for \ i = 1{:}n \tag{7}$$

where $y_t^o$ is the recorded measurement at time $t$ and $\Sigma_t^{y^o y^o}$ is the error covariance matrix of
the measurements.
3. **Generate simulations using prior parameters**. The prior parameters from Step 1, $\theta_t^{i-}$ and
updated states from the previous time, $x_{t-1}^{i+}$ are forced through the model equations to
generate an ensemble of model simulations of states ($\hat{x}_t^i$) and outputs ($\hat{y}_t^i$):

$$\hat{x}_t^i = f\left(x_{t-1}^{i+}, \theta_t^{i-}, u_t^i\right) \ for \ i = 1{:}n \tag{8}$$

$$\hat{y}_t^i = h\left(\hat{x}_t^i, \theta_t^{i-}\right) \ for \ i = 1{:}n \tag{9}$$

4. **Perform the Kalman update of parameters.** Parameters are updated using the Kalman
update equation and the prior parameter and simulated output ensemble from Step 1 and 3:

$$\theta_t^{i+} = \theta_t^{i-} + \mathbf{K}_t^\theta\left(y_t^i - \hat{y}_t^i\right) \ for \ i = 1{:}n \tag{10}$$

$$\mathbf{K}_t^\theta = \Sigma_t^{\theta\hat{y}}\left[\Sigma_t^{\hat{y}\hat{y}} + \Sigma_t^{y^o y^o}\right]^{-1} \tag{11}$$

where $\Sigma_t^{\theta\hat{y}}$ is a matrix of the sample cross covariance between errors in parameters $\theta_t^{i-}$ and
simulated output $\hat{y}_t^i$ ; and $\Sigma_t^{\hat{y}\hat{y}}$ is the sample error covariance matrix of the simulated output:

$$\Sigma_t^{\theta\hat{y}} = \frac{1}{n-1}\sum_{i=1}^{n}\left(\theta_t^{i-} - \overline{\theta_t^-}\right)\left(\hat{y}_t^i - \overline{\hat{y}_t}\right)^{\mathrm{T}} \tag{12}$$

$$\Sigma_t^{\hat{y}\hat{y}} = \frac{1}{n-1}\sum_{i=1}^{n}\left(\hat{y}_t^i - \overline{\hat{y}_t}\right)\left(\hat{y}_t^i - \overline{\hat{y}_t}\right)^{\mathrm{T}} \tag{13}$$

5. ***Generate simulations using updated parameters.*** Step 3 is repeated with the updated
parameter ensemble $\boldsymbol{\theta}_t^{i+}$ to generate the prior ensemble of model simulations of states ($\boldsymbol{x}_t^{i-}$)
and outputs ($\widetilde{\boldsymbol{y}}_t^i$):

$$\boldsymbol{x}_t^{i-} = f\left(\boldsymbol{x}_{t-1}^{i+}, \boldsymbol{\theta}_t^{i+}, \boldsymbol{u}_t^i\right) \ for \ i = 1{:}n \tag{14}$$

$$\widetilde{\boldsymbol{y}}_t^i = h\left(\boldsymbol{x}_t^{i-}, \boldsymbol{\theta}_t^{i+}\right) \ for \ i = 1{:}n \tag{15}$$

6. ***Perform the Kalman update of states and outputs***. Use the Kalman update equation for
correlated measurement and process noise (equations 16 to 19) and the simulated state
($\boldsymbol{x}_t^{i-}$) and output ($\widetilde{\boldsymbol{y}}_t^i$) ensembles from Step 5 to update them.  Since the measurements have
already been used to generate $\widetilde{\boldsymbol{y}}_t^i$, the errors in model simulations and measurements are
now correlated.  The standard Kalman update equation (as in the form of equations 10 and
11) can no longer be used as it relies on the assumption that errors in measurements and
model simulations are independent.

$$\boldsymbol{x}_t^{i+} = \boldsymbol{x}_t^{i-} + \mathbf{K}_t^x\left(\boldsymbol{y}_t^i - \widetilde{\boldsymbol{y}}_t^i\right) \ for \ i = 1{:}n \tag{16}$$

$$\mathbf{K}_t^x = \left[\boldsymbol{\Sigma}_t^{x\widetilde{y}} + \boldsymbol{\Sigma}_t^{\varepsilon_x y^o}\right]\left[\boldsymbol{\Sigma}_t^{\widetilde{y}\widetilde{y}} + \boldsymbol{\Sigma}_t^{\varepsilon_{\widetilde{y}} y^o} + \left(\boldsymbol{\Sigma}_t^{\varepsilon_{\widetilde{y}} y^o}\right)^{\mathrm{T}} + \boldsymbol{\Sigma}_t^{y^o y^o}\right]^{-1} \tag{17}$$

$$\boldsymbol{\varepsilon}_{x_t}^i = \boldsymbol{x}_t^{i-} - \widehat{\boldsymbol{x}}_t^i \tag{18}$$

$$\boldsymbol{\varepsilon}_{\widetilde{y}_t}^i = \widetilde{\boldsymbol{y}}_t^i - \widehat{\boldsymbol{y}}_t^i \tag{19}$$

where $\boldsymbol{\Sigma}_t^{x\widetilde{y}}$ is a matrix of the sample cross covariance between simulated states $\left\{\boldsymbol{x}_t^{i-}\right\}_{i=1:n}$
and outputs $\left\{\widetilde{\boldsymbol{y}}_t^i\right\}_{i=1:n}$ from Step 5; $\boldsymbol{\Sigma}_t^{\varepsilon_x y^o}$ represents the sample covariance between
$\left\{\boldsymbol{\varepsilon}_{x_t}^i\right\}_{i=1:n}$ and the observations; and $\boldsymbol{\Sigma}_t^{\varepsilon_{\widetilde{y}} y^o}$ represents the sample covariance between the
$\left\{\boldsymbol{\varepsilon}_{\widetilde{y}_t}^i\right\}_{i=1:n}$ and the observations.
The above algorithm specifies the updating of states and parameters at any given time, based on
available observations.  This allows one to retrospectively estimate time variations in model
parameters, as well as provide one time step ahead forecasts of states & outputs (as per equations 8
and 9).  Forecasts at longer time horizons (i.e. longer than one time step ahead) would be made by
generating prior parameters and states as detailed in Steps 1 to 3, although the local linear
extrapolations are only valid close to the current time point.
**3.2.2. Application to the Nammuc Catchment**
Joint state and parameter estimation was undertaken for the Nammuc Catchment over the period
1980 to 2004 by assimilating streamflow observations into the HyMOD and HBV models at a daily
time step.  Additionally, simulations using the time invariant parameters obtained from calibration
over the period 1973-1979 were generated for 1980 to 2004, for comparison.  Estimating a large
number of parameters from limited data is problematic in that the system is highly under-
determined, making it difficult to ensure the estimated parameters are meaningful.  Given the fairly
low parameter dimensionality of HyMOD, all model parameters were allowed to vary in time whilst
for HBV we applied the Sobol method to identify the most sensitive parameters to be included in the
time varying parameter estimation.  The Sobol method is a global sensitivity analysis method based
on variance decomposition. It identifies the partial variance contribution of each parameter to the
total variance of the hydrological model output [see for example *Saltelli et al.,* 2008, *Nossent et al.*
*2011*]. The method, implemented through the SAFE toolbox [*Pianosi et al.,* 2015], found the $lp$ and
$Maxbas$ parameters to be the least sensitive and least important in defining variations to catchment
hydrology (see Table 3). These were held fixed ($lp$ = 1 and $Maxbas$ = 1 day) in the following analysis.
Note that although the $hl1$ parameter was found to have low sensitivity, it was retained as a time
varying parameter due to its conceptual importance in separating interflow and near surface flow
(refer Figure 3).

Unbiased normally distributed ensembles of the parameters and states are required to initialise the
LL Dual EnKF.  Initial parameter ensembles were generated by sampling from a Gaussian distribution
with mean equal to the calibrated parameters over the pre-change period and variance estimated
from parameter sets with similar objective function values.  Parameter sets with similar objective
function values were obtained when using different starting points to the optimization algorithm
during the model calibration stage.  Initial state ensembles were also sampled from normal
distributions with mean equal to the simulated state at the end of the calibration period.  An
ensemble size of 100 members was adopted and assumed sufficiently large based on the findings of
*Moradkhani et al.* [2005] and *Aksoy et al.* [2006].  Due to the stochastic-dynamic nature of the
method, ensemble statistics were calculated over 20 separate realisations of the LL Dual EnKF.  The
prior parameter generating method described in Step 1 of Section 3.2 requires specification of the
tuning parameter $s^2$ to define the variance of the perturbations.  This was tuned by selecting the $s^2$
value that optimized the quality of forecast streamflow over the calibration period. Forecast quality
was assessed using the logarithmic score (LS) [*Good,* 1952] of background streamflow predictions
($\tilde{y}_t^i$) using updated parameters (equation 15), which was averaged over the calibration period of
length *T*:

$$\overline{LS} = \sum_{t=1}^{T} LS_t \tag{20}$$

$$LS_t = \log\left(f(y = y_t^o)\right) \tag{21}$$

where $f(y)$ is the probability density function of the background streamflow predictions
(represented by the empirical pdf of the sample points $\{\tilde{y}_t^i\}_{i=1:n}$); and $y_t^o$ is the measurement of the
system outputs.  The $s^2$ value that gave the largest $\overline{LS}$ was adopted for the assimilation period.  The
maximum allowable daily rate of change in the ensemble mean was based on assuming a linear rate
of change within the entire feasible parameter space over a three year period.

As detailed in Section 3.2, observation and forcing uncertainty is considered by perturbing
measurements with random noise.  Here streamflow errors were assumed to be zero-mean normally
distributed (truncated to ensure positivity) and heteroscedastic.  The variance is defined as a
proportion of the observed streamflow, to reflect the fact that larger flows tend to have greater
errors than low flows:

$$y_t^i \sim TN(y_t^o, d.y_t^o) \ for \ i = 1:n \qquad (22)$$

where TN indicates the truncated normal distribution to ensure positive flows and $d$ = 0.1.  A
multiplier of 0.1 was chosen based on estimates adopted for similar gauges in hydrologic DA studies
[e.g. *Clark et al.,* 2008; *Weerts & Serafy,* 2006; *Xie et al.,* 2014].

Several studies have noted that a major source of rainfall uncertainty arises from scaling point
rainfall to the catchment scale [*Villarini & Krajewski*, 2008; *McMillan et al.*, 2011] and that
multiplicative errors models are suited to describing such errors [e.g. *Kavetski et al.*, 2006]. Rainfall
uncertainties were therefore described using unbiased, lognormally distributed multipliers:

$$P_t^i = P_t.M^i \qquad (23)$$
$$M^i \sim LN(m, v) \ and \ X^i = \log(M^i) \sim N(\mu, \sigma^2) \quad for \ i = 1:n \qquad (24)$$

where $P_t$ is the measured rainfall at time $t$; $m$ and $v$ are the mean and variance of the lognormally
distributed rainfall multipliers $M$ respectively; and $\mu$ and $\sigma^2$ are the mean and variance of the
normally distributed logarithm of the rainfall multipliers $M$.  For unbiased perturbations, we let $m$ =
1.  The variance of the rainfall multipliers ($v$) was estimated by considering upper and lower bound
error estimates in the Thiessen weights assigned to the four rainfall stations (see Section 2.1 for
calculation of catchment averaged rainfall, $P_t$).  The resulting upper and lower bound catchment
averaged rainfall data were then used to estimate error parameters due to spatial variation in
rainfall:

$$v = e^{(2\mu + \sigma^2)}.\left(e^{\sigma^2} - 1\right) \qquad (25)$$
$$\sigma^2 = \widehat{\sigma^2} = var\left(\log\left[\frac{P_{upper,10}}{P_{lower,10}}\right]\right) \qquad (26)$$
$$\mu = \log(m) - \frac{\sigma^2}{2} = -\frac{\sigma^2}{2} \qquad (27)$$

where $P_{upper,10}$ indicates catchment averaged rainfall data estimated using the upper bound
Thiessen weights with daily depth greater than 10mm (similar for $P_{lower,10}$).  A 10mm rainfall depth
threshold was chosen to avoid large rainfall fractions due to small rainfall depths.  $\widehat{\sigma^2}$  was found to
be 0.05 in this case study.  Similarly, we assume the dominant source of uncertainty in temperature
data arises from spatial variation.  Differences in temperature records at Lai Chau and Quynh Nhai
(only available gauges with temperature records) were analysed and found to be approximately
normally distributed with sample mean 0.2 deg C and variance of 1.4 deg C.  A perturbed
temperature ensemble was then generated according to equation 28:

$$T_t^i \sim TN\left(T_t^{avg}, 1.4\right) \ \ for \ \ i = 1{:}n$$  ( 28 )

where $T_t^{avg}$ represents catchment averaged temperature data (see Section 2.1).  Note that
perturbations were taken to be unbiased (zero mean) as the sample mean of the differences in the
temperature records was close to zero.  The same perturbed input and observation sequences were
used for the HyMOD and HBV runs for the sake of comparison. A summary of the values adopted for
the various components of the Locally Linear Dual EnKF for each model is provided in Table 4 and
Table 5.

## 4.  Results and Discussion

Temporal variations in the estimated parameter distributions from the LL Dual EnKF are evident for
both models (see Figure 4 and 5).  In the case of the HBV model, changes at an inter-annual time
scale are evident for the $perc$ and $\beta$ (see Figure 4).  The decrease in the $\beta$ parameter means that a
greater proportion of rainfall is converted to runoff (i.e. more water entering the shallow layer
storage).  Additionally, the increase in the $perc$ parameter means that a greater volume of water is
made available for baseflow generation.  These changes correspond with the observed increase in
the annual runoff coefficient (Figure 2) and increase in baseflow volume (as discussed in Section 2.2).
From an algorithm perspective, these parameters are most strongly correlated with streamflow (as
well as the most sensitive, see Table 3), meaning that they will receive the greatest proportional
updates.   Similar parameter adjustments are seen for HyMOD, at least at a qualitative level (see
Figure 5). The sharp increase in the $b$ parameter during the post-change period means that a greater
volume of water is available for routing (as larger $b$ values mean that a smaller proportion of the
catchment has deep soil storage capacity) and the downward inter-annual trend in $\alpha$ means that a
greater portion of excess runoff is routed through the baseflow store. Intra-annual variations in
updated model parameters for both HyMOD and HBV are also apparent (refer Figure 4 and Figure 5).
This is due to the inability of a single parameter distribution to accurately model both wet and dry
season flows. Such variations were not observed when using the time varying parameter framework
for small deforested catchments (< 350ha) [see *Pathiraja et al.,* 2016b]. The comparatively less clear
parameter changes for the Nammuc catchment are due to a combination of the increased difficulty
in accurately modelling the hydrologic response (even in pre-change conditions) and due to the
relatively more subtle and gradual changes to land cover. Nonetheless, the method is shown to
generate a temporally varying structure that is conceptually representative of the observed changes.

Despite the overall correspondence between changes to model parameters and observed
streamflow, a closer examination shows that the hydrologic model structure is critical in determining
whether the time varying parameter models accurately reflect changes in all aspects of the
hydrologic response (not just total streamflow). In order to examine the impact of parameter
variations on the model dynamics, we generated model simulations with the time varying parameter
ensemble from the LL Dual EnKF, but without state updating (hereafter referred to as TVP-HBV and
TVP-HyMOD). Streamflow predictions from the LL Dual EnKF (i.e. with state and parameter updating)
for both the HyMOD and HBV are generally of similar quality and superior to those from the
respective time invariant parameter models that have been calibrated on pre-change data (1975-
1979), although a slight bias in baseflow predictions from HyMOD is evident (see for example Figure
6). The Nash Sutcliffe Efficiency of one step ahead streamflow predictions over the period 1980 –
2004 from the LL Dual EnKF is 0.87 when using HyMOD or HBV, compared to 0.76 and 0.72 for the
respective time invariant parameter models evaluated over the same period. However, differences
in predictions from TVP-HBV and TVP-HyMOD are more striking due to the lack of state updating.
Figure 7 shows annual statistics of simulated streamflow from the TVP-HBV and TVP-HyMOD models
and observed runoff.  The TVP-HBV gives direct runoff and baseflow predictions that are consistent
with runoff observations, meaning that the parameter adjustments reflect the observed changes in
the runoff response.  This however is not the case for the TVP-HyMOD. The annual runoff coefficient
and annual direct runoff coefficient are severely under-estimated in the post-change period by the
TVP-HyMOD, whilst the Annual Baseflow Index has an increasing trend of magnitude far greater than
observed (Figure 7c).  All three quantities on the other hand are well represented by the TVP-HBV
(Figure 7).  Similar conclusions can be drawn from Figure 8, which shows the results of a Moving
Average Shifting Horizon (MASH) analysis (see Section 2.2) on total and direct runoff (observed and
simulated).  Observed increases in January to April flows (see Figure 8a) and wet season direct flows
(July to September) (see Figure 8e) are well represented by the TVP-HBV but not TVP-HyMOD.

The reason for the differences in performance between the TVP-HBV and TVP-HyMOD lies in the
structure of the hydrologic model.  The TVP-HyMOD is incapable of representing the observed
increase in annual runoff/direct runoff coefficient due to the increased baseflow during dry periods,
despite having an Annual Baseflow Index far greater than the observed.  This occurs due to an
inability to generate flow volume during periods of no rain.   In joint state-parameter updating using
HyMOD, underestimated runoff predictions during dry periods lead to adjustments to the $k_s$ and $\alpha$
parameters to increase baseflow depth (since these are the only parameters that are associated to
an active store).  Unlike HBV, HyMOD has no continuous supply of water to the routing stores (i.e.
the quick flow and slow flow stores) during recession periods (which typically have extended periods
of no rainfall, so that $V$ in Figure 3 is zero).  This means that $k_s$ and $\alpha$ are updated to extreme values
to compensate for the volumetric shortfall.  The HBV structure, on the other hand, has a continuous
percolation of water into the deep layer store even during periods of no rain (so long as the shallow
water store is non-empty).  In summary, the HyMOD model structure is poorly suited to simulating
streamflow dynamics in post-change conditions, although it gave reasonable simulations in pre-
change conditions.  This highlights that need to select a sufficiently flexible model structure prior to
undertaking forecasting/predictive modelling using the time varying parameter approach.  In
particular, the model structure must be capable of effectively simulating all potential future
catchment conditions.

Having established that the TVP-HBV provided a good representation of the observed streamflow
dynamics, we used a modelling approach to determine whether the observed changes were solely
driven by forcings and which (if any) components of runoff were also affected by land use change.  A
resampled rainfall and temperature time series was generated by sampling the data without
replacement across years for each day (for instance rainfall and temperature for $1^{st}$ January 1990 is
found by randomly sampling from all records on $1^{st}$ January).  This maintains the intra-annual (e.g.
seasonal) variability but destroys any inter-annual trends in the meteorological data.  Streamflow
simulations were then generated using this resampled meteorological sequence as inputs to the TVP-
HBV (i.e. without state updating).  If the resulting streamflow simulations do not reproduce the
observed changes to streamflow dynamics, then this indicates that changes to meteorological
forcings are the main contributor.  However, if it is able to at least partially (or fully) reproduce the
observed streamflow changes, this means that land cover changes are impacting catchment
hydrology (but potentially in addition to forcing changes, due to the presence of ecosystem
feedbacks). Figure 8d&h show the results of a MASH undertaken on the resulting simulations of total
and direct runoff using the resampled forcing time series and TVP-HBV model.  Observed increases in
baseflow during the January – April period (see Figure 8a) and increases in direct runoff in the June –
September period (see Figure 8e) are reproduced.  The magnitude of increase in direct runoff in July
is slightly lower, indicating the potential for some climatic influences also. This is consistent with
findings from the Mann-Kendall test which identified a statistically significant increase in July rainfall
(see Section 2.2).  Overall however, these results lend further weight to the conclusion that land
cover change has impacted the hydrologic regime of the Nammuc catchment.  These results also
demonstrate that parameter changes correspond to actual changes in catchment hydrology, and are
not just random fluctuations that reproduce the observed streamflow statistics only when the
observed forcing time series is used.
## 5. Conclusions
As our anthropogenic footprint expands, it will become increasingly important to develop modelling
methodologies that are capable of handling changing catchment conditions.  Previous work proposed
the use of models whose parameters vary with time in response to signals of change in observations.
The so-called Locally Linear Dual EnKF time varying parameter estimation algorithm [*Pathiraja et al.,*
2016a] was applied to 2 sets of small (< 350 ha) paired experimental catchments with deforestation
occurring under experimental conditions (rapid clearing of 100% and 50% of land surface) [*Pathiraja*
*et al.,* 2016b].  Here we demonstrate the efficacy of the method for a larger catchment experiencing
more realistic land cover change, whilst also investigating the importance of the chosen model
structure in ensuring the success of the time varying parameter estimation method.  We also
demonstrate that the time varying parameter framework can be used in a retrospective fashion to
determine whether land cover changes (and not just meteorological factors) contribute to the
observed hydrologic changes.

Experiments were undertaken on the Nammuc catchment (2880 km$^2$) in Vietnam, which experienced
a relatively gradual conversion from forest to cropland over a number of years (cropland increased
from roughly 23% of the catchment between 1981 and 1994 to 52% by 2000).  Changes to the
hydrologic regime after the mid-1990s were detected and attributed mostly to an increase in
baseflow volume.  Application of the LL Dual EnKF with two conceptual models (HBV and HyMOD)
showed that the time varying parameter framework with state updating improved streamflow
prediction in post-change conditions compared to the time invariant parameter case.  However,
baseflow predictions from the LL Dual EnKF with HBV were generally superior to the HyMOD case
which tended to have a slight negative bias. It was found that the structure (i.e. model equations) of
HyMOD was unsuited to representing the modified baseflow conditions, resulting in extreme and
unrealistic time varying parameter estimates. This work shows that the chosen model is critical for
ensuring the time varying parameter framework successfully models streamflow in unknown future
land cover conditions, particularly when used in a real time forecasting mode. Appropriate model
selection can be a difficult task due to the significant uncertainty associated with future land use
change, and can be even more problematic when multiple models have similar performance in pre-
change conditions (as was the case in this study). One possible way to ensure success of the time
varying parameter approach is to use models whose fundamental equations explicitly represent key
physical processes (for instance, modelling sub-surface flow using Richard's equation with hydraulic
conductivity allowed to vary with time). In this way, time variations in model parameters would
more closely reflect changes to physiographic properties, rather than also having to account for
missing processes. The drawback of such physically based models is that they are generally data
intensive, both in generating model simulations (i.e. detailed inputs) and specifying parameters.
Additionally, it may be necessary to reduce the dimensionality of the time varying parameter vector
by keeping less sensitive model parameters fixed in order to make the estimation problem tractable.
Models of intermediate complexity that have explicit process descriptions may be the most
promising, although this also remains to be demonstrated.
## 6. Acknowledgements
This study was funded by the Australian Research Council as part of the Discovery Project
DP140102394. Dr. Marshall is additionally supported through a Future Fellowship FT120100269. The
research of Dr. Pathiraja has been partially funded by Deutsche Forschungsgemeinschaft (DFG)
through the grant CRC 1294 "Data Assimilation."

The data used in this paper were collected under the project IMRR (Integrated and sustainable water
Management of Red Thai Binh Rivers System in changing climate), funded by the Italian Ministry of
Foreign Affairs (Delibera n. 142 del 8 Novembre 2010).  We greatly acknowledge Dr. Andrea
Castelletti for provision of data and for discussions on this work.

Data utilized in this study can be made available from the authors upon request.

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

**Tables**

| | Pre 1994 | Post 1994 |
|---|---|---|
| *Land Use* | | |
| **Evergreen Forest (including evergreen needle and evergreen leaf) (%)** | 77% | 48% |
| **Cropland (%)** | 23% | 52% |
| *Hydro-Meteorological Properties* | | |
| **Mean Annual Rainfall (mm)** | 1630 | 1660 |
| **Mean Annual Runoff (mm)** | 838 | 1190 |
| **Mean Annual Runoff Coefficient** | 0.5 | 0.7 |
| **Mean Annual PET (mm)** | 1300 | 1300 |
| **Estimated Mean Annual BFI** | 0.33 | 0.39 |

**Table 1 Study catchment properties**








| | HYMOD | HBV |
|---|---|---|
| NSE [ ] | **0.77** | 0.75 |
| *Peak flows (q > 5mm/d)* | | |
| MAE [mm/d] | 3.11 | **2.85** |
| RMSE [mm/d] | **4.55** | 4.72 |
| *Medium flows (1 mm/d <= q <= 5mm/d)* | | |
| MAE [mm/d] | **0.66** | 0.80 |
| RMSE [mm/d] | **0.86** | 1.09 |
| *Low flows (q < 1mm/d)* | | |
| MAE [mm/d] | 0.35 | **0.20** |
| RMSE [mm/d] | 0.42 | **0.34** |

**Table 2 Model performance in pre-change conditions used for calibration (1975 – 1979). Bold face**
**numbers correspond to the model with superior performance for the particular metric. NSE = Nash**
**Sutcliffe Efficiency; MAE = Mean Absolute Error; RMSE = Root Mean Squared Error.**





| | Sensitivity Index |
|---|---|
| *hl1* | 0.10 |
| *lp* | 0.12 |
| *Maxbas* | 0.14 |
| *fcap* | 0.18 |
| *K0* | 0.23 |
| *K2* | 0.23 |
| *K1* | 0.38 |
| *beta* | 0.41 |
| *perc* | 0.47 |

**Table 3 Variance Based Sensitivity Analysis Results for HBV parameters: first order sensitivity index**
**representing the contribution of varying a single parameter to the variance of the model output.**
**Lower values indicate lower sensitivity.**





| Parameters | | | | | | |
|---|---|---|---|---|---|---|
| | Description | Units | Initial Sampling Distribution | Feasible Range | $s^2$ | Max allowable daily rate of change ($m_{max}$) |
| $\beta$ | Soil Moisture exponent | [ ] | $N(2, 0.1)$ | $0 - 7$ | 0.003 | $1.8 \times 10^{-3}$ |
| $fcap$ | Maximum soil moisture store depth | [mm] | $N(467, 10)$ | $10 - 2000$ | 0.003 | 0.4 |
| $hl1$ | Threshold for generation of near surface flow | [mm] | $N(120, 10)$ | $0 - 400$ | 0.003 | 0.1 |
| $K0$ | Near Surface Flow Routing Coefficient | [ ] | $N(0.3, 0.005)$ | $0.0625 - 1$ | 0.003 | $2 \times 10^{-4}$ |
| $K1$ | Interflow Routing Coefficient | [ ] | $N(0.09, 5 \times 10^{-4})$ | $0.02 - 0.1$ | 0.003 | $9 \times 10^{-6}$ |
| $perc$ | Percolation rate | [mm/d] | $N(1.3, 10^{-4})$ | $0 - 3$ | 0.003 | $10^{-3}$ |
| $K2$ | Baseflow Routing Coefficient | [ ] | $N(0.01, 10^{-6})$ | $5 \times 10^{-5} - 0.02$ | 0.003 | $9 \times 10^{-6}$ |
| States | | | | | | |
| $sowat$ | Soil Moisture Store | [mm] | $N(0,1)$ | $(0, fcap)$ | | |
| $stw1$ | Shallow Layer Store | [mm] | $N(0,1)$ | $(0, \infty)$ | | |
| $stw2$ | Deep Layer Store | [mm] | $N(0,0.1)$ | $(0, \infty)$ | | |

**Table 4 Locally Linear EnKF inputs for the HBV model case**


| | | | | | | Max allowable |
|---|---|---|---|---|---|---|
| | | | | | | daily rate of |
| | | | | | | change ($m_{max}$) |

| | Description | Units | Initial Sampling Distribution | Feasible Range | $s^2$ | Max allowable daily rate of change ($m_{max}$) |
|---|---|---|---|---|---|---|
| **Parameters** | | | | | | |
| $b$ | Pareto-distributed soil storage shape parameter | [ ] | $N(0.37, 10^{-4})$ | $0 - 0.3$ | 0.004 | $3\times10^{-4}$ |
| $c_{max}$ | Maximum point soil storage depth | [mm] | $N(651, 10)$ | $300 - 1500$ | 0.004 | 0.3 |
| $k_q$ | Quick flow Routing Coefficient | [ ] | $N(0.6, 5\times10^{-4})$ | $0.55 - 0.99$ | 0.018 | $3\times10^{-4}$ |
| $k_s$ | Slow flow Routing Coefficient | [ ] | $N(0.04, 5\times10^{-4})$ | $0.001 - 0.54$ | 0.018 | $4\times10^{-5}$ |
| $\alpha$ | Excess Runoff Splitting Parameter | [ ] | $N(0.47, 5\times10^{-4})$ | $0.001 - 0.99$ | 0.018 | $4\times10^{-4}$ |
| **States** | | | | | | |
| $S$ | Soil Store | [mm] | $N(180, 0.1*180)$ | $(0, S_{max} = \frac{bc_{min} + c_{max}}{b+1})$ | | |
| $S_{q1,2,3}$ | Quick Flow Stores | [mm] | $N(0,1)$ | $(0, \infty)$ | | |
| $S_s$ | Slow Flow Store | [mm] | $N(0,1)$ | $(0, \infty)$ | | |

**Table 5 Locally Linear EnKF inputs for the HYMOD model case**





**Figures**

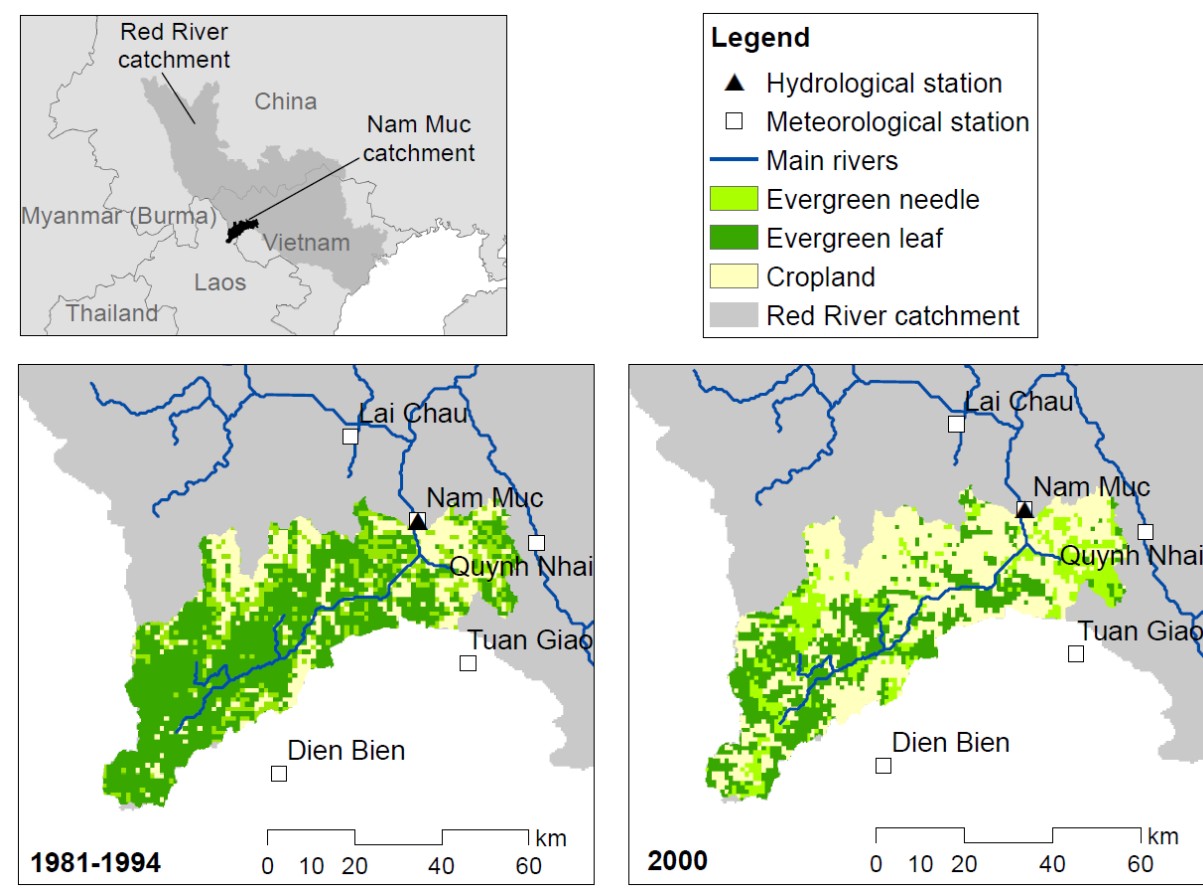


**Figure 1 Study Catchment showing gauges and changes in land cover over time.**





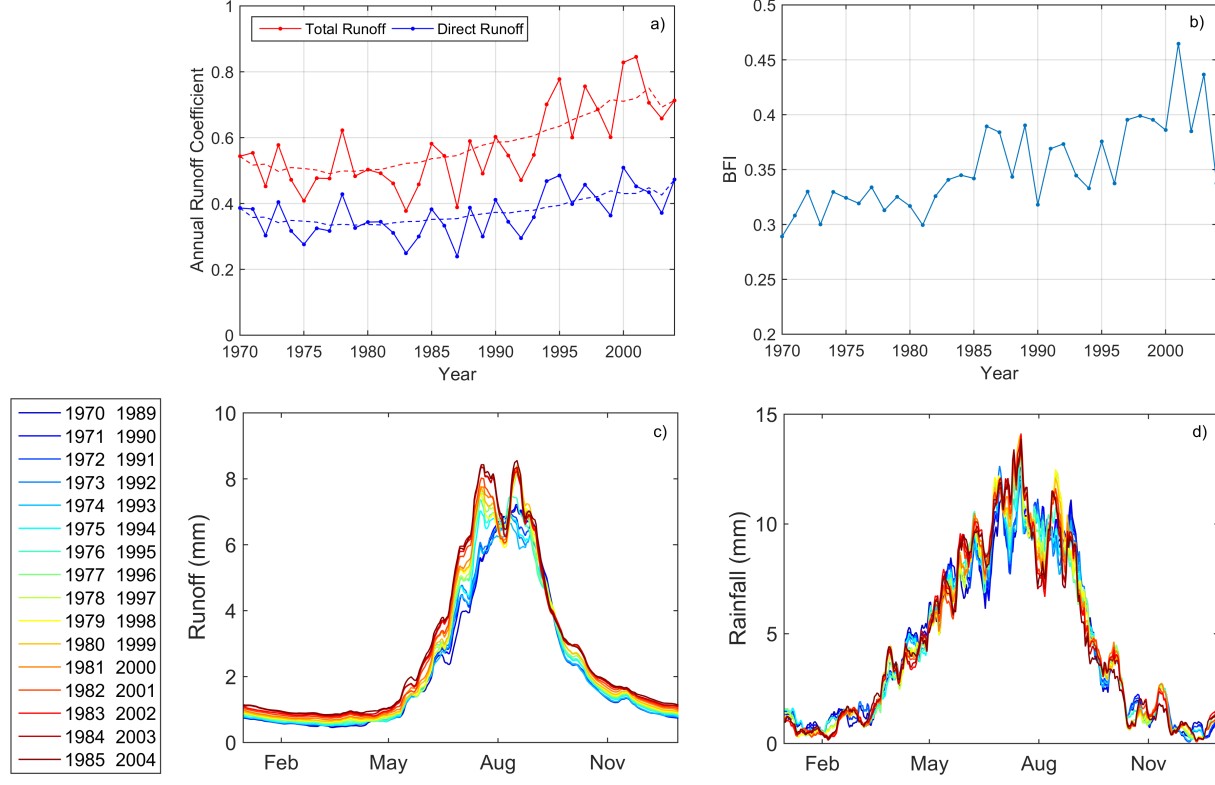




**Figure 2 Impact of land use change on observed streamflow: a) Annual Runoff Coefficient, b) Annual Baseflow Index (BFI), c) Moving Average Shifting Horizon (MASH) results for total observed runoff, d) MASH for observed rainfall.**










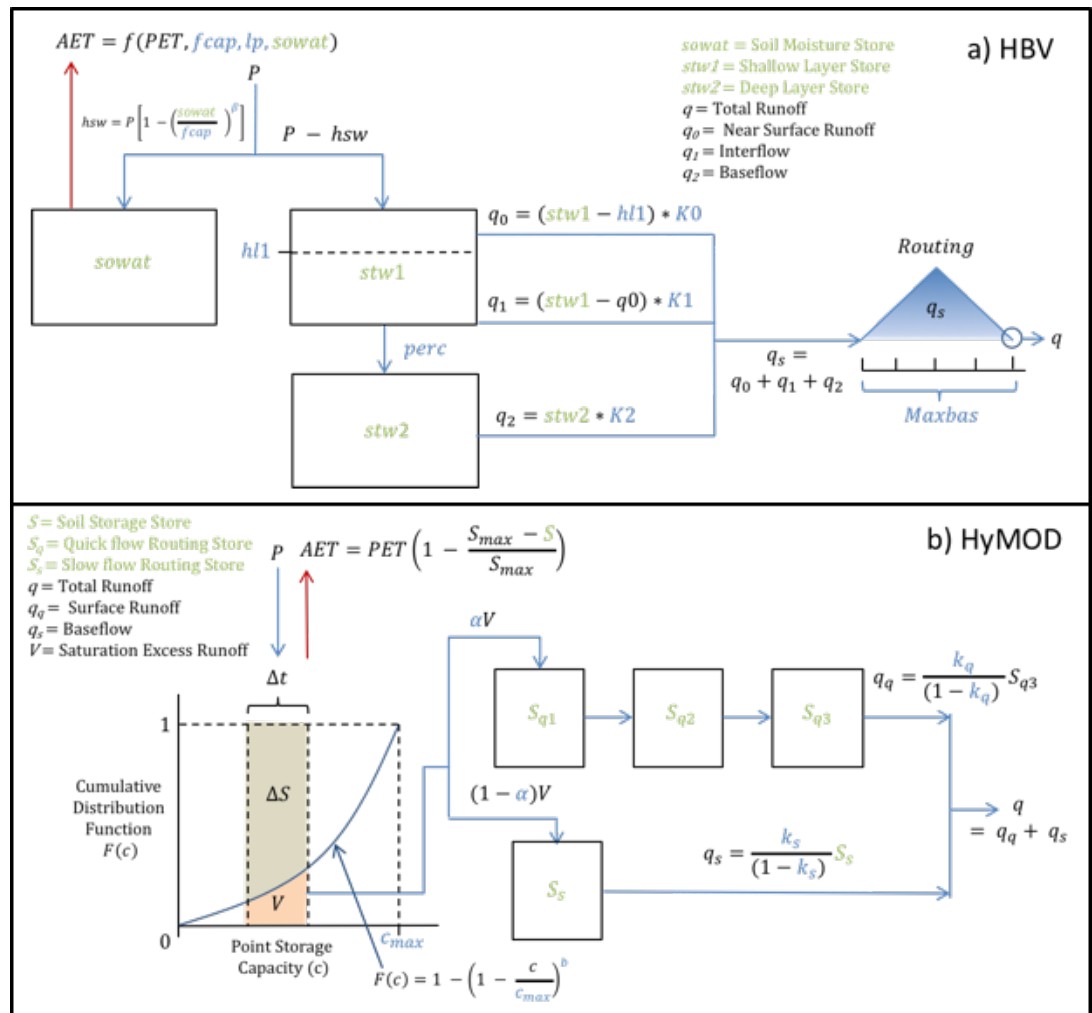

**Figure 3 Schematic of the models used in this study: a) HBV and b) HyMOD. Parameters are shown in blue and states are shown in green.**

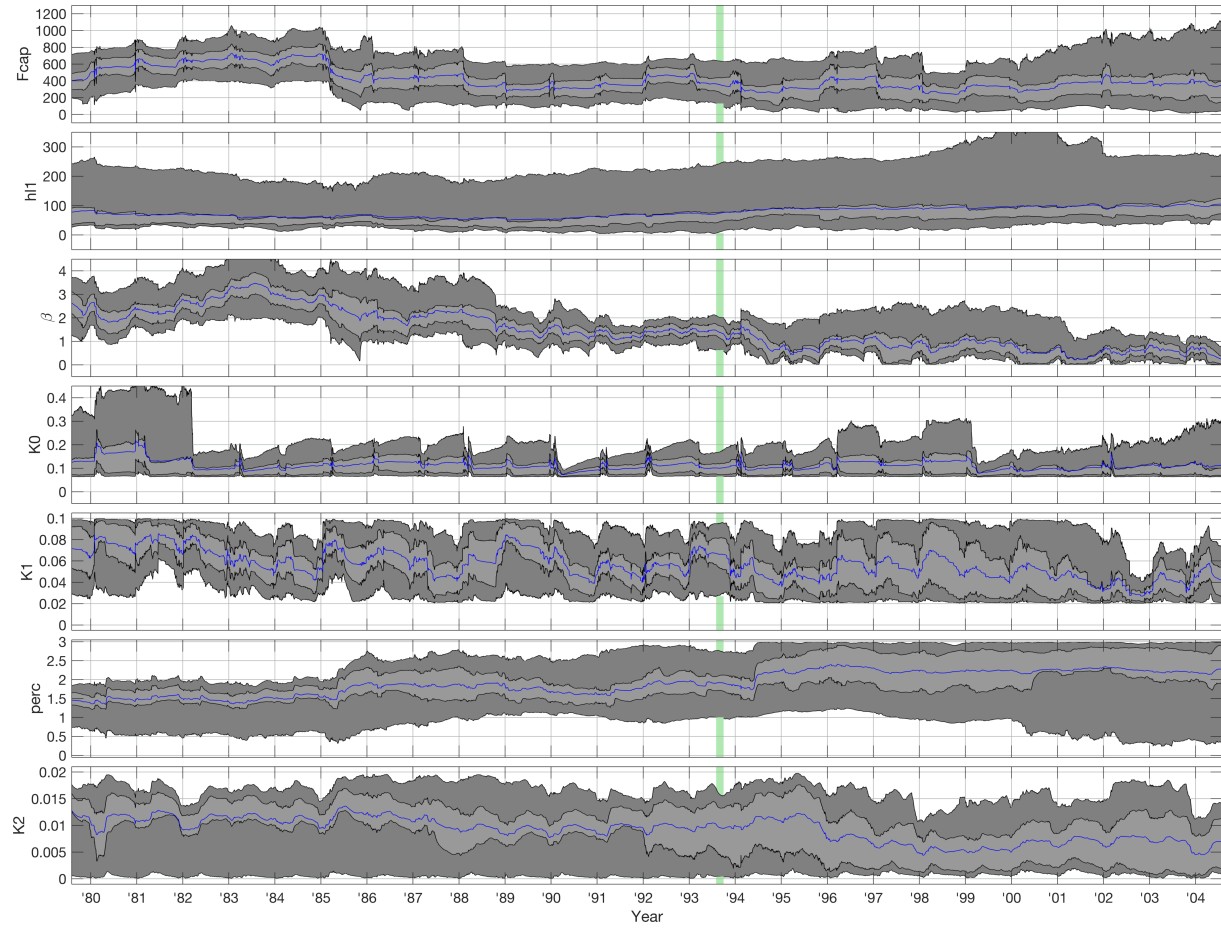

**Figure 4 Parameter Trajectories using the HBV model.  The dark grey shaded areas indicate the middle 90% of the ensemble, bounded by the 5th and 95th percentiles.  The light grey shaded areas indicate the middle 50% of the ensemble, bounded by the 25th and 75th percentiles.  The ensemble mean is indicated by the blue line.  The vertical green panel indicates the assumed time period of rapid deforestation.**

819

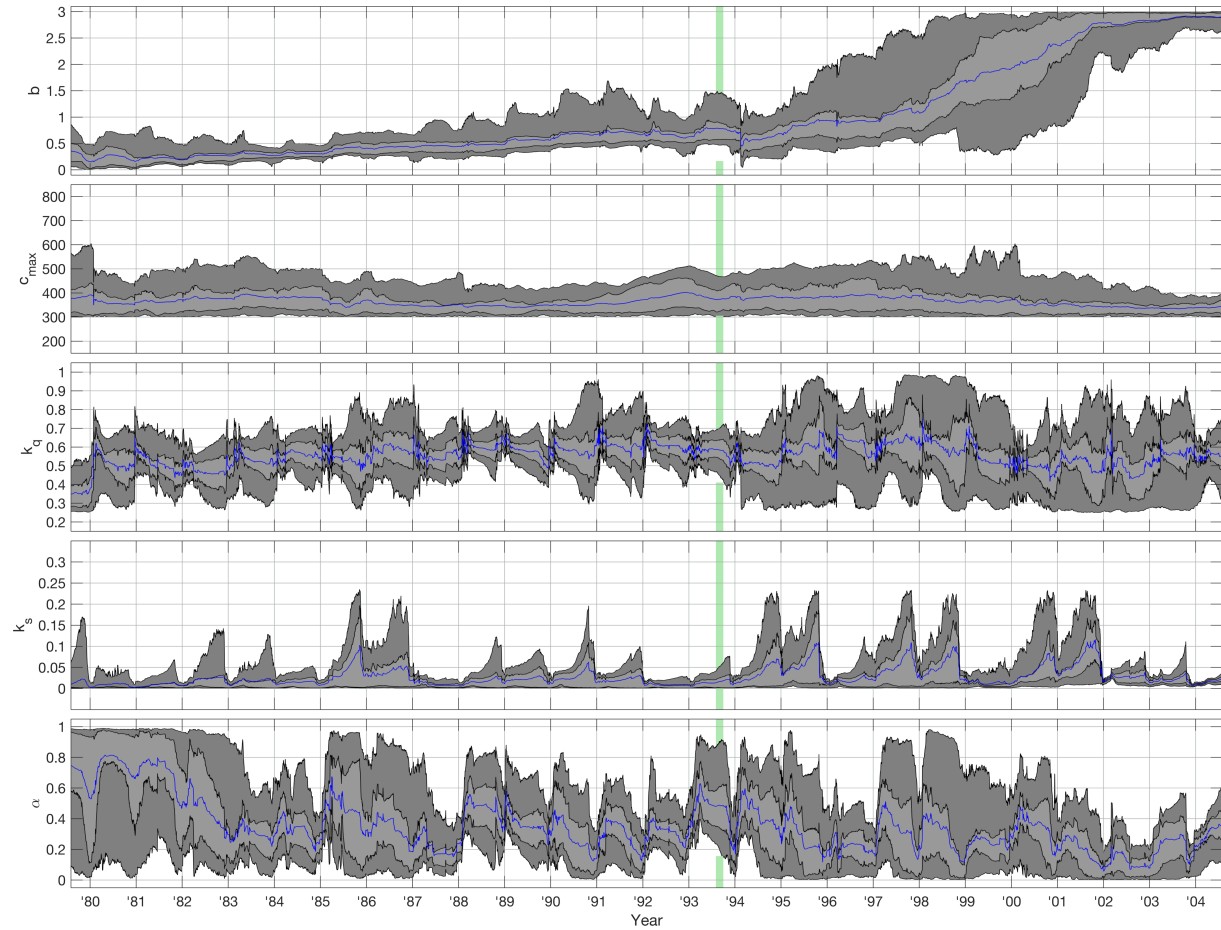

820

**Figure 5 Parameter Trajectories using the HyMOD model. The dark grey shaded areas indicate the middle 90% of the ensemble, bounded by the 5th and 95th percentiles. The light grey shaded areas indicate the middle 50% of the ensemble, bounded by the 25th and 75th percentiles. The ensemble mean is indicated by the blue line. The vertical green panel indicates the assumed time period of rapid deforestation.**


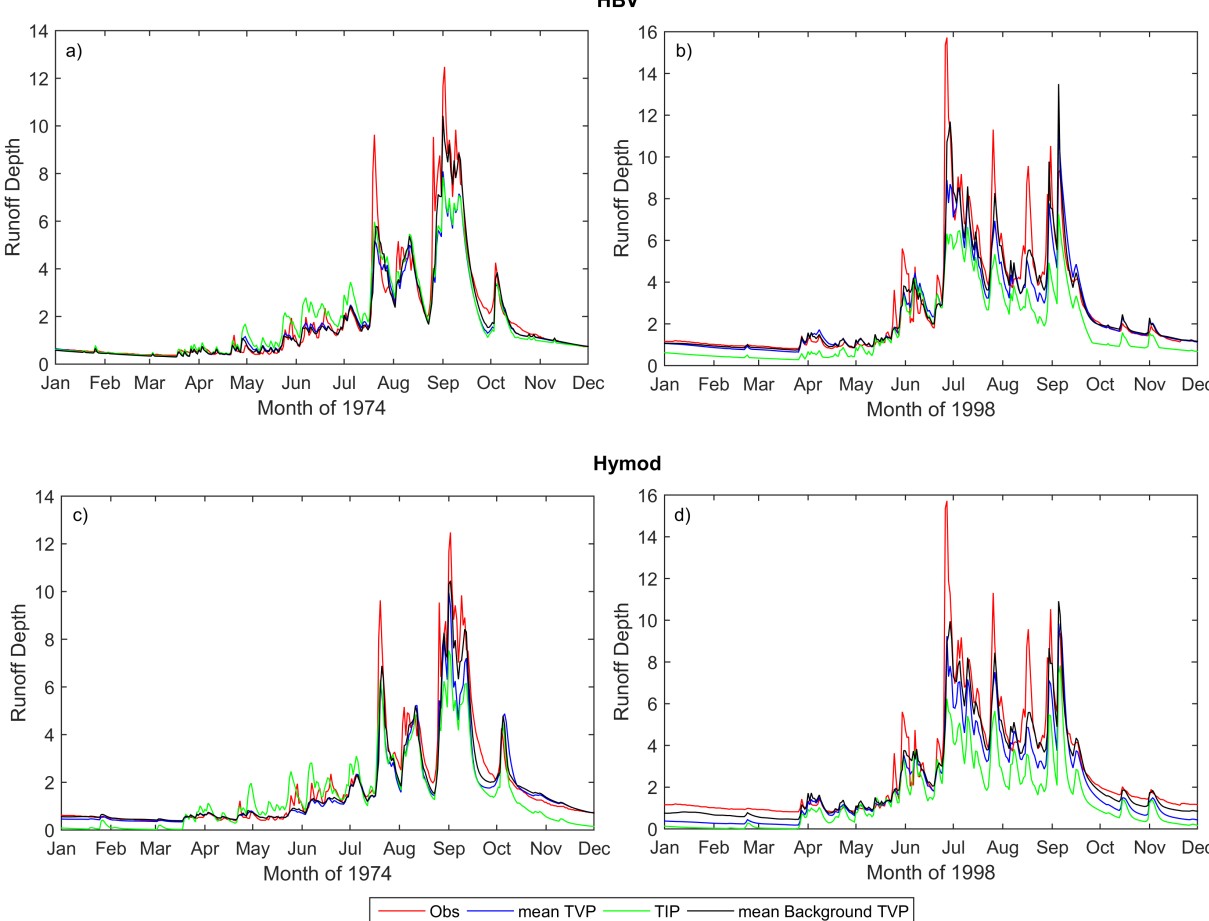

**Figure 6 Representative Hydrographs of background streamflow from the LL Dual EnKF (black line), Time varying parameter model with no state updating (blue line), time invariant parameter model with no DA (green line) and observed streamflow (red line). Results for HBV are shown in the top row and HyMOD in the bottom row. A pre-change year (1974) is shown on the left and a post change year (1998) on the right.**

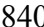

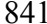

Figure 7 Influence of time varying parameters on model output (i.e. without state updating) summarized in terms of the Annual Runoff Coefficient (top row), Annual Direct Runoff Coefficient (second row) and Annual Baseflow Index (BFI) (third row). Results for HyMOD are shown in the first column, HBV are shown in the second column.

848

849

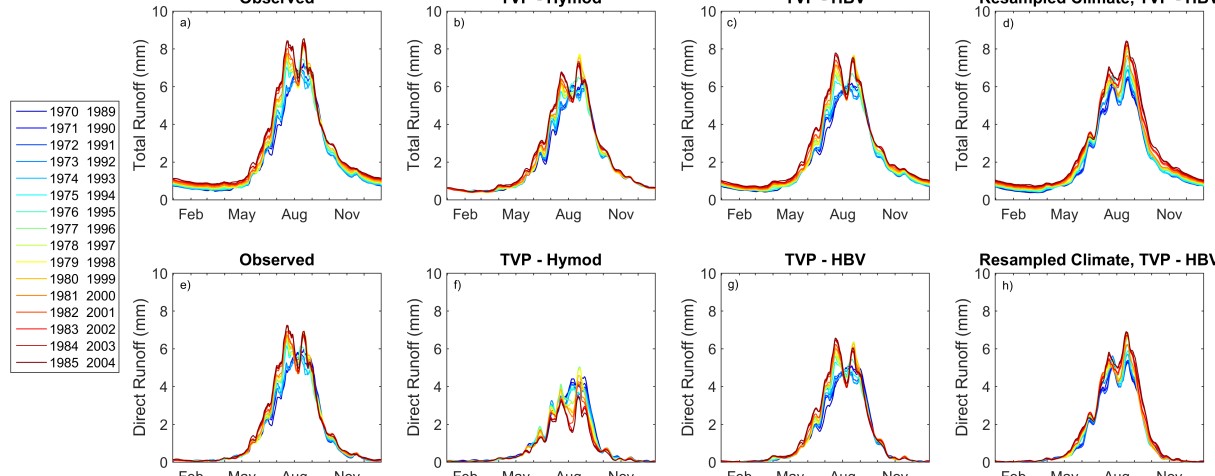

850

**Figure 8 Moving Average Shifting Horizon (MASH) results for observed streamflow (first column), simulated streamflow from time varying parameter model (without state DA) for HYMOD (2nd column), HBV (third column), resampled climate HBV (fourth column). These are split into total runoff (first row) and direct runoff or surface runoff (2nd row).**

855