# Peer review of "change: the importance of model structure"

_Hydrology and Earth System Sciences, 2017_

## Author Comment (AC1) · 4 Aug 2017

| | Parameters | | | | | |
|---|---|---|---|---|---|---|
| | Description | Units | Initial Sampling Distribution | Feasible Range | Initial $s^2$ (VVM) | Max allowable daily rate of change (LL) |
| $b$ | Pareto-distributed soil storage shape parameter | [ ] | $N(0.37, 10^{-4})$ | $0 – 0.3$ | 0.004 | $3\times10^{-4}$ |
| $c_{max}$ | Maximum point soil storage depth | [mm] | $N(651, 10)$ | $300 – 1500$ | 0.004 | 0.3 |
| $k_s$ | Surface Runoff Routing Coefficient | [ ] | $N(0.6, 5\times10^{-4})$ | $0.55 – 0.99$ | 0.018 | $3\times10^{-4}$ |
| $k_b$ | Groundwater Routing Coefficient | [ ] | $N(0.04, 5\times10^{-4})$ | $0.001 – 0.54$ | 0.018 | $4\times10^{-5}$ |
| $\alpha$ | Excess Runoff Splitting Parameter | [ ] | $N(0.47, 5\times10^{-4})$ | $0.001 – 0.99$ | 0.018 | $4\times10^{-4}$ |
| | States | | | | | |
| $S$ | Soil Store | [mm] | $N(180, 0.1*180)$ | $(0, S_{max} = \frac{bc_{min}+ c_{max}}{b+1})$ | | |
| $S_{q1,2,3}$ | Quick Flow Stores | [mm] | $N(0,1)$ | $(0, \infty)$ | | |
| $S_s$ | Slow Flow Store | [mm] | $N(0,1)$ | $(0, \infty)$ | | |

**Table 5: Locally Linear EnKF inputs for the HYMOD model case**

---

## Referee Comment (RC1) · Anonymous Referee #1 · 23 Aug 2017

The authors are performing data assimilation (DA), for a dynamical hydrological model with time-varying parameters. I fully agree that, in hydrology, time-varying parameters very often lead to a more realistic description of reality than constant ones, with the caveat that a good stochastic model for their dynamics is used. That being said, I have serious doubts about the validity of the method used in this paper.

First of all, there is no clear separation between the hydrological model assumptions and the numerical method that is used for DA. The model assumptions should not only comprise the deterministic hydrological equations and the observational error assumptions, but here also a precise definition of the assumed stochastic dynamics of the

model parameters. These assumptions have nothing to do with DA, but are part of our prior knowledge about the system. Together with prior probability distributions, e.g. for initial conditions, and measured data they completely define the posterior as well as predictive distributions of states and parameters. All DA methods must lead to the same distributions and the choice is only a matter of computational efficiency. If this separation is not clearly made, there is a risk that inference (here DA) and prediction is done under different model assumptions, which would be inconsistent and lead to a loss of interpretability of the results. I've been trying to do this separation and unveil the assumptions behind the dynamics of the parameters from step 1 of the numerical DA algorithm, but it is not obvious to me what these assumptions are. Without constraining data (i.e. in a predictive mode), what would be the dynamics of the parameter distribution?

Putting aside my concerns about these model assumptions, I have even more serious concerns about the chosen DA method, which seems to violate the assumptions behind Kalman filters in at least two ways: (i) Kalman filters assume normality of the distribution of the augmented state (incl. parameters). Is there any reason to believe that the non-linearity of the chosen hydrological model is weak enough for this assumption to be approximately valid? Given the dimensionality of the model and the data, respectively, I'm almost certain that it will be grossly violated. (ii) Updating the states, based on prior predictions that have been made with parameters that have already been updated seems to use the data twice. This again seems to violate model assumptions, or in other words, I have no idea what the model assumptions are, for which the proposed method is a valid DA.

---

## Referee Comment (RC2) · Anonymous Referee #2 · 6 Sep 2017

This manuscript tested a time varying model parameter framework at a river basin under significant land cover changes in the last few decades. The employed framework is based on Locally Linear Dual EnKF proposed by the authors in the previous studies and is applied to the two conceptual hydrologic models (i.e. HBV and HyMOD).

The manuscript shows interesting result and well written in general. However, I have concerns regarding practical applications of the tested approach and the objectives of this research from the following aspects.

1. In the abstract, the authors stated "rapid land use change impacts on catchment hydrology" and "therefore modeling methodology of such change is important" in the

first and second sentences. First of all, please clarify "for what purpose" such modeling representing the land cover change is thought to be needed. This could be, for example, estimating future water resources under further land cover and climate changes or identifying the physical mechanisms of the impact of land cover change on hydrology. Please explain in the introduction for what objective the authors think the modeling with time varying parameter is necessary.

2. Based on the firs point, please explain how the applied framework with the EnKF can achieve the objectives. Obviously the presented approach requires a full set of input and output to estimate the parameter changes. Suppose that this approach now successfully estimates the time-varying parameters in the HBV model, how can this information be useful for water management, given already land cover change has happened and streamflow change has been already detected in the actual catchment.

3. Related to the above point, please state clearly the main objective of this research in the introduction. Is the main objective here to test the time varying model parameter framework in the data limited catchment? In such case, what is the criteria to conclude the objective has been achieved. The EnKF may show the parameter changes, but is it enough to validate the method? Or is the main objective here to compare the two model structures?

With the above main review comments, I have the followings minor comments.

1. Abstract L18 and L57: "it serves as an effective tool for separating the influence of climatic and land use change": is this really true? As a result of the EnKF, it is possible that the both effects of land cover and climate changes may be reflected in the wrong way. Given such an ill-identified potential, please explain the logic and actual steps to distinguish the impacts of the two changes.

2. P7 L122 Subsection of "2.1 " may be eliminated because no "2.2" exists.

3. P10 L206 Is the covariance matrix (sigma) also updated in the sequential Kalman

Filter algorithm? Please clarify this part and show the equation if it is also updated.

4. P10 L218 The same comment is also applied to the Kalman gain of the model parameters.

5. P12 L261 How did you select the tuning parameter s2? The used values in this manuscript should be shown. Table 4 shows "Initial s2 (VVM)" which confuses me because I thought that s2 were set as constant value for each parameter.

6. P17 "a MASH undertaken..." Please add a brief explanation of the MASH approach.

---

## Author Comment (AC2) · 15 Sep 2017

**Response to reviewer 1**

Responses to the comments are shown in black and proposed changes to the

manuscript are shown in blue.

The authors are performing data assimilation (DA), for a dynamical hydrological model with time-varying parameters. I fully agree that, in hydrology, time-varying parameters very often lead to a more realistic description of reality than constant ones, with the caveat that a good stochastic model for their dynamics is used. That being said, I have serious doubts about the validity of the method used in this paper.

We thank the reviewer for their time. Please see below our responses to the comments.

First of all, there is no clear separation between the hydrological model assumptions and the numerical method that is used for DA. The model assumptions should not only comprise the deterministic hydrological equations and the observational error assumptions, but here also a precise definition of the assumed stochastic dynamics of the model parameters. These assumptions have nothing to do with DA, but are part of our prior knowledge about the system. Together with prior probability distributions, e.g. for initial conditions, and measured data they completely define the posterior as well as predictive distributions of states and parameters.

We respectfully note that the assumed stochastic dynamics of the model parameters were detailed in Step 1 of the algorithm in Section 3.2. The assumed parameter dynamics are a Gaussian random walk with time varying mean and variance. This is a reasonable assumption for cases where no prior knowledge of the parameter dynamics (or changes to catchment properties) is available. The prior mean is assumed to evolve locally linearly in time, such that it is estimated by the drift rate from the previous two time steps (as indicated in Line 204). This is shown in Figure 1. The prior variance is assumed to be also time dependent, and based on the variance at the previous time (as indicated in Lines 205-207). The statistics of the prior parameter ensemble are also updated sequentially by the DA algorithm based on assimilated observations.

We note that these details are not the main focus of the paper, as they have already been investigated and discussed in detail in Pathiraja et al. (2016). We have provided references to these publications in the manuscript (see Lines 184, 191-192). However, we will provide additional discussion in the revised manuscript, as detailed below.

We propose to add the following to Step 1 of Section 3.2 so that the assumed stochastic dynamics are clearer: "The parameters are assumed to evolve through a Gaussian random walk with time varying mean and variance. The prior mean is assumed to evolve locally linearly in time, such that it is estimated by the drift rate from the previous two time steps. The prior variance is based on the variance of the update from the previous time." This discussion would be inserted in Line 202 before "The prior (or background)..."

Figure 1. Schematic showing how the mean of the prior parameter ensemble  $\overline{\theta}_{t+1}^-$  is proposed at a given time *t*+1. [*taken from* Pathiraja et al. (2016)]

All DA methods must lead to the same distributions and the choice is only a matter of computational efficiency. If this separation is not clearly made, there is a risk that inference (here DA) and prediction is done under different model assumptions, which would be inconsistent and lead to a loss of interpretability of the results. I've been trying to do this separation and unveil the assumptions behind the dynamics of the parameters from step 1 of the numerical DA algorithm, but it is not obvious to me what these assumptions are. Without constraining data (i.e. in a predictive mode), what would be the dynamics of the parameter distribution?

DA methods in general do not lead to the same posterior distributions, particularly when applied to complex non-linear/non-Gaussian problems. This is even when one is specifying the prior, initial conditions and observation error in exactly the same way. One only has to look at the plethora of existing DA algorithms to appreciate this (Ensemble Kalman Filter, Ensemble Transform Kalman Filter, Ensemble Adjustment Kalman Filter, Particle Filter-SIR, Particle Filter-MCMC, Auxiliary Particle Filter, just to name a few). They do not differ solely in terms of computational efficiency (e.g. Arulampalam et al., 2002; Liu et al., 2007).

The assumed parameter stochastic dynamics were discussed in the response to the previous comment; these apply in predictive mode also. In this case, the proposed local linear extrapolation is only valid close to the current time point. This is acceptable given that the Locally Linear Dual EnKF is designed for short term predictive modelling. This will be emphasised in the revised manuscript.

We propose to insert the following sentence at Line 64: "It can be used to either retrospectively estimate time variations in model parameters or for short-term predictive modelling."

Putting aside my concerns about these model assumptions, I have even more serious concerns about the chosen DA method, which seems to violate the assumptions behind Kalman filters in at least two ways:

(i) Kalman filters assume normality of the distribution of the augmented state (incl. parameters). Is there any reason to believe that the non-linearity of the chosen hydrological model is weak enough for this assumption to be approximately valid? Given the dimensionality of the model and the data, respectively, I'm almost certain that it will be grossly violated. We note that the method relies on an Ensemble Kalman Filter, not a standard Kalman Filter. Ensemble Kalman Filters were specifically designed to handle the non-linear/non-Gaussian case, although they are sub-optimal in that only the mean and covariance are considered in the update. Nevertheless, they are a practical alternative to Sequential Monte Carlo methods (which approximate the full Bayesian posterior), as these methods suffer from several practical implementation issues. We also note that Ensemble Kalman Filters have been used with considerable success in a wide range of complex non-linear/non-Gaussian problems (see for Reichle et al., 2002; Gu et al., 2005; Komma et al., 2008; Sun et al., 2009; Xu et al., 2016).

**(ii) Updating the states, based on prior predictions that have been made with parameters that have already been updated seems to use the data twice. This again seems to violate model assumptions, or in other words, I have no idea what the model assumptions are, for which the proposed method is a valid DA.**

In regards to the second issue, in Pathiraja et al. (2016), we demonstrated that the dual update filter accounts for correlations between observation and process noise associated with the use of a dual update (see Step 6 in Section 3.2). Full details of the dual update procedure were provided in Pathiraja et al. (2016). The purpose of this manuscript is not to revisit details of this method, references to the relevant publications were provided in Lines 184 and 191-192 of this manuscript.

**References**

- Arulampalam, M. S., S. Maskell, N. Gordon, and T. Clapp (2002), A tutorial on particle filters for online nonlinear/non-Gaussian Bayesian tracking, *IEEE Trans. Signal Process.*, 50(2), 174–188, doi:10.1109/78.978374.
- Gu, Y., and D. S. Oliver (2005), History matching of the PUNQ-S3 reservoir model using the ensemble Kalman filter, *SPE J.*, *10*(2), 217–224, doi:10.2118/89942-PA.
- Komma, J., G. Blöschl, and C. Reszler (2008), Soil moisture updating by Ensemble Kalman Filtering in real-time flood forecasting, *J. Hydrol.*, *357*(3–4), 228–242, doi:10.1016/j.jhydrol.2008.05.020.
- Liu, Y., and H. V. Gupta (2007), Uncertainty in hydrologic modeling: Toward an integrated data assimilation framework, *Water Resour. Res.*, 43(7), doi:10.1029/2006WR005756.
- Pathiraja, S., L. Marshall, A. Sharma, and H. Moradkhani (2016), Hydrologic modeling in dynamic catchments: A data assimilation approach, *Water Resour. Res.*, 52, 3350–3372, doi:10.1002/2015WR017192.
- Reichle, R. H., D. B. McLaughlin, and D. Entekhabi (2002), Hydrologic Data Assimilation with the Ensemble Kalman Filter, *Am. Meteorol. Soc. Mon. Weather Rev.*, *130*(1), 103–114, doi:10.1175/1520-0493(2002)130<0103:HDAWTE>2.0.CO;2.
- Sun, A. Y., A. Morris, and S. Mohanty (2009), Comparison of deterministic ensemble Kalman filters for assimilating hydrogeological data, *Adv. Water Resour.*, 32(2), 280–292, doi:10.1016/j.advwatres.2008.11.006.
- Xu, T., and J. . Gomez-Hernandez (2016), Joint identification of contaminant source location, initial release time, and initial solute concentration in an aquifer via ensemble kalman filtering, *Water Resour. Res.*, 600–612, doi:10.1002/2015WR018249.

---

## Author Comment (AC3) · 15 Sep 2017

**Response to reviewer 2**

Responses to the comments are shown in black and proposed changes to the manuscript are shown in blue.

*This manuscript tested a time varying model parameter framework at a river basin under significant land cover changes in the last few decades. The employed framework is based on Locally Linear Dual EnKF proposed by the authors in the previous studies and is applied to the two conceptual hydrologic models (i.e. HBV and HyMOD).*

*The manuscript shows interesting result and well written in general. However, I have concerns regarding practical applications of the tested approach and the objectives of this research from the following aspects.*

We thank the reviewer for their time and appreciate the overall positive assessment of the manuscript. We address concerns regarding practical applications below.

*1. In the abstract, the authors stated "rapid land use change impacts on catchment hydrology" and "therefore modeling methodology of such change is important" in the first and second sentences. First of all, please clarify "for what purpose" such modelling representing the land cover change is thought to be needed. This could be, for example, estimating future water resources under further land cover and climate changes or identifying the physical mechanisms of the impact of land cover change on hydrology. Please explain in the introduction for what objective the authors think the modeling with time varying parameter is necessary.*

We have stated in the introduction that the purpose of the time varying parameter framework is to provide accurate hydrologic predictions even when catchment conditions are changing (see Lines 31-32). We also discussed that the time varying parameter framework could be used to isolate whether changes to streamflow dynamics are driven by climatic or land cover changes (see Lines 57 – 59). We propose to add additional discussion as outlined in the response to the next comment.

*2. Based on the firs point, please explain how the applied framework with the EnKF can achieve the objectives. Obviously the presented approach requires a full set of input and output to estimate the parameter changes. Suppose that this approach now successfully estimates the time-varying parameters in the HBV model, how can this information be useful for water management, given already land cover change has happened and streamflow change has been already detected in the actual catchment.*

The Locally Linear Dual EnKF can be useful for retrospective analysis of variations to model parameters. From a prediction perspective, the framework is useful for short term predictions (or forecasts) of hydrologic variables in catchments that are undergoing change. The reason it can be used in this context is because parameters are being updated on-the-fly, in response to the observations that are being assimilated (as stated in Lines 62 – 64). It cannot be used for water management over a long time horizon, as this requires explicit prior information about future land use change. The purpose of this approach is to infer changes to catchment

properties from hydrologic observations, at the time scale of the observation frequency. Therefore, it is not suited to water management purposes over a long time horizon, as this requires explicit prior information about future land use change.

We propose to insert the following sentence at Line 64, before "The method was applied…":

"Its purpose is to infer changes to catchment properties (e.g. land cover change) from hydrologic observations, without prior knowledge of such changes, at the time scale of the observation frequency. It can therefore be used to either retrospectively estimate time variations in model parameters or for short-term predictive modelling."

***3. Related to the above point, please state clearly the main objective of this research in the introduction. Is the main objective here to test the time varying model parameter framework in the data limited catchment? In such case, what is the criteria to conclude the objective has been achieved. The EnKF may show the parameter changes, but is it enough to validate the method? Or is the main objective here to compare the two model structures?***

The main objectives here are:
1) To investigate the efficacy of the time varying parameter method in a medium sized catchment with realistic land cover change. This goes beyond previous work which focused on small experimental catchments that had drastic land use changes that are easier to infer from streamflow; and
2) To highlight the importance of the chosen model structure in ensuring the success of the time varying parameter method.

The above statements were provided in the Abstract (see Lines 9 – 11), Introduction (see Lines 61 – 76) and Conclusion (Lines 380 – 390).

In regards to validation of the time varying parameter method, its efficacy was assessed based on the ability to represent all aspects of the streamflow hydrograph. In Section 4, we showed that time varying parameter HBV model provided a good representation of baseflow, total and direct runoff (see Figure 7 and 8, and discussion in Lines 329-361). We also showed that the time varying parameter HyMOD model did not represent all aspects of the hydrograph. This showed the importance of the chosen model structure in determining the success of the time varying parameter method.

***With the above main review comments, I have the followings minor comments.***

***1. Abstract L18 and L57: "it serves as an effective tool for separating the influence of climatic and land use change": is this really true? As a result of the EnKF, it is possible that the both effects of land cover and climate changes may be reflected in the wrong way. Given such an ill-identified potential, please explain the logic and actual steps to distinguish the impacts of the two changes.***

The logic here is that if the observed changes to streamflow are driven by changes to meteorological forcings, then there should be no changes to the model parameters (assuming the prior parameterisation and model represent the catchment properties and processes well). In this case, any changes to meteorological forcings would be translated through to simulated

streamflow, so that the prior streamflow closely matches the observed streamflow. This means there will be a small to negligible update of the parameters (see equation 1 of the manuscript, where the term $(y_t^i - \hat{y}_t^i)$ will be small). We also provided additional analysis with a resampled forcing time series and the TVP-HBV model to demonstrate that climatic changes were not the main driver of observed changes to streamflow (see lines 363 – 378).

**2. P7 L122 Subsection of "2.1 " may be eliminated because no "2.2" exists.**

This change will be incorporated in the revised manuscript.

**3. P10 L206 Is the covariance matrix (sigma) also updated in the sequential Kalman Filter algorithm? Please clarify this part and show the equation if it is also updated.**

The term $\Sigma_{t-1}^{\theta}$ is simply the sample covariance of the updated (or posterior) parameter ensemble $\{\theta_{t-1}^{i+}\}_{i=1:n}$ at time *t-1*. In Ensemble Kalman Filtering, the covariance matrix is not explicitly updated, but is replaced by the sample covariance of the updated parameter/state members. The equation for updating the members is given in equations 1 and 3 of the manuscript.

We will insert the word "sample" before "covariance matrix" in line 206.

**4. P10 L218 The same comment is also applied to the Kalman gain of the model parameters.**

We are unclear as to the meaning of this comment. The Kalman gain is what is used to update the model parameters.

As with the previous comment, we can insert the word "sample" before "cross covariance" in Line 222 and before "error covariance matrix" in Line 224.

**5. P12 L261 How did you select the tuning parameter s2? The used values in this manuscript should be shown. Table 4 shows "Initial s2 (VVM)" which confuses me because I thought that s2 were set as constant value for each parameter.**

An explanation of how the $s^2$ parameter was tuned was provided in Section 3.2.1 (please see Lines 261 – 263). The values used are shown in Table 4, and are constant values.

The column header in Table 4 will be changed from "Initial s2 (VVM)" to just "s2." We apologise for the error here.

**6. P17 "a MASH undertaken..." Please add a brief explanation of the MASH approach.**

We propose to add the following discussion at Line 140 before "A steady increase in …":

"The MASH tool can be used to qualitatively assess interannual variations in the seasonal pattern of a variable. It works by calculating a statistic of the data (e.g. mean) over the same block of days in each year."

---

## Author Response (AR1)

**Response to the Editor**

Dear Dr. Hrachowitz,

Thank you for evaluating our manuscript and review responses with a keen eye and providing constructive comments. We have modified our manuscript to address all review comments. In particular, we have significantly expanded the discussion of the method used in the manuscript, paying special attention to comments from Reviewer 1 (see Section 3.2.1). It is hoped that this will serve as a self-contained explanation of the method. We have also provided additional discussion on DA methods so as to explain the reasons for adopting an EnKF based approach (see Lines 233-241) along with its main limitation (i.e. the sub-optimal update, see Line 236).

We have also modified the introduction so that the science questions of interest are more clearly stated (see Lines 89-113). These are two specific issues that arise when applying the time varying parameter method to realistic study settings, which, to the best of our knowledge, has not been previously investigated. We summarise our approach to investigating these issues in Lines 115-127. More details of the practical applications of the time varying method have also been provided, as requested by Reviewer 2 (see Lines 33-40, 75-84).

In regards to the comment on separating meteorological and land use change contributions: we acknowledge that the interpretations from our analysis were oversimplified in our initial response. Our intent is to demonstrate that the time varying parameter approach can be used to show that observed changes to streamflow are not solely a result of changes in the meteorological patterns (as per the analysis at the end of Section 4). We have provided additional discussion to make clear how the analysis should be interpreted, taking into consideration helpful comments from both Reviewer 2 and the Editor (see Lines 479-484, 482-484). Specifically, it cannot be used to precisely separate out the contributions of forcing and land use change; the analysis can only be used to show that either 1) meteorological changes are the main driver of streamflow changes or 2) land cover changes are contributing to hydrologic change (but potentially in addition to meteorological changes due to the presence of ecosystem feedbacks, as noted by the Editor).

Full details of our revisions can be found in the response to reviewer documents, along with responses to all comments.

Thank you for your time and consideration.

Best Regards,

Sahani Pathiraja
Daniela Anghileri
Paolo Burlando
Ashish Sharma
Lucy Marshall
Hamid Moradkhani

**Response to reviewer 1**

*Please note all line references refer to the revised document with tracked changes.*

***The authors are performing data assimilation (DA), for a dynamical hydrological model with time-varying parameters. I fully agree that, in hydrology, time-varying parameters very often lead to a more realistic description of reality than constant ones, with the caveat that a good stochastic model for their dynamics is used. That being said, I have serious doubts about the validity of the method used in this paper.***

We thank the reviewer for their time. Please see below our responses to specific comments.

***First of all, there is no clear separation between the hydrological model assumptions and the numerical method that is used for DA. The model assumptions should not only comprise the deterministic hydrological equations and the observational error assumptions, but here also a precise definition of the assumed stochastic dynamics of the model parameters. These assumptions have nothing to do with DA, but are part of our prior knowledge about the system. Together with prior probability distributions, e.g. for initial conditions, and measured data they completely define the posterior as well as predictive distributions of states and parameters.***

We have provided additional explanation on how the prior parameters are generated in the revised manuscript (see Lines 268-286). This is undertaken in Step 1 of the algorithm in Section 3.2.1. The assumed parameter dynamics are a Gaussian random walk with time varying mean and variance. This is a reasonable assumption for cases where no prior knowledge of the parameter dynamics (or changes to catchment properties) is available. The prior mean is assumed to evolve locally linearly in time, such that it is estimated by the drift rate from the previous two time steps (eqn 1 and eqns 3-5). The prior variance is assumed to be also time dependent, and based on the variance at the previous time (eqn 2). The statistics of the prior parameter ensemble are updated sequentially by the DA algorithm based on assimilated observations.

We have also provided more detailed explanations of the other steps in the algorithm (see Section 3.2.1). It is hoped that this will serve as a self-contained explanation of the method.

***All DA methods must lead to the same distributions and the choice is only a matter of computational efficiency. If this separation is not clearly made, there is a risk that inference (here DA) and prediction is done under different model assumptions, which would be inconsistent and lead to a loss of interpretability of the results. I've been trying to do this separation and unveil the assumptions behind the dynamics of the parameters from step 1 of the numerical DA algorithm, but it is not obvious to me what these assumptions are. Without constraining data (i.e. in a predictive mode), what would be the dynamics of the parameter distribution?***

We have provided additional discussion to make clear the DA method used, associated limitations, and why it has been adopted (see Lines 233-243). DA methods in general do not lead to the same posterior distributions, particularly when applied to complex non-linear/non-Gaussian problems. This is even when one is specifying the prior, initial conditions and observation error in exactly the same way. For instance, Kalman Filter based methods (e.g. Ensemble Kalman Filter, Ensemble Transform Kalman Filter, Ensemble Adjustment Kalman Filter) consider only the mean and covariance in generating the update (see Line 236), whilst Particle Filter methods (e.g. Particle Filter-SIR, Particle Filter-MCMC, Auxiliary Particle Filter) aim to propagate the full probability density through time (see Line 239-240). They do not differ solely in terms of computational efficiency (e.g. Arulampalam et al., 2002; Liu et al., 2007). An

EnKF based approach has been adopted because it is a practical alternative to Particle Filter/Sequential Monte Carlo methods with demonstrated success in many hydrologic applications (see Lines 237-239).

The assumed parameter stochastic dynamics were discussed in the response to the previous comment; these apply in predictive/forecast mode also. In this case, the proposed local linear extrapolation is only valid close to the current time point. This is acceptable given that the Locally Linear Dual EnKF is designed for short term predictive modelling/forecasting. This has been made clear in Lines 327-332. The intended use of the algorithm has also been explicitly described in Lines 75-81.

***Putting aside my concerns about these model assumptions, I have even more serious concerns about the chosen DA method, which seems to violate the assumptions behind Kalman filters in at least two ways:***

***(i) Kalman filters assume normality of the distribution of the augmented state (incl. parameters). Is there any reason to believe that the non-linearity of the chosen hydrological model is weak enough for this assumption to be approximately valid? Given the dimensionality of the model and the data, respectively, I'm almost certain that it will be grossly violated.***

We note that the method relies on an Ensemble Kalman Filter, not a standard Kalman Filter. Ensemble Kalman Filters were specifically designed to handle the non-linear/non-Gaussian case. This means that they at least partially consider non-Gaussianity in state variables, as the full non-linearity of the forward model is considered when calculating prior variables. However, they are sub-optimal in that only the mean and covariance are considered in calculating the update or posterior. Nevertheless, they are a practical alternative to Sequential Monte Carlo methods (which approximate the full Bayesian posterior), as these methods suffer from several practical implementation issues even in moderate dimensional problems (such as degeneracy and sample impoverishment). We also note that Ensemble Kalman Filters have been used with considerable success in a wide range of complex non-linear/non-Gaussian problems (see for Reichle et al., 2002; Gu et al., 2005; Komma et al., 2008; Sun et al., 2009; Xu et al., 2016).

We have provided further discussion in the manuscript in relation to the above (please see lines 232-243).

***(ii) Updating the states, based on prior predictions that have been made with parameters that have already been updated seems to use the data twice. This again seems to violate model assumptions, or in other words, I have no idea what the model assumptions are, for which the proposed method is a valid DA.***

Please note that in Step 6 of the algorithm (Section 3.2.1), the Kalman Update equation for correlated process and measurement noise is used instead of the standard Kalman update equation. This is used precisely to account for the fact that the observations have already been used in generating the prior states. A detailed derivation of these equations (i.e. equations 15 to 18 in the revised manuscript) can be found in the Appendix B of Pathiraja et al. (2016). We have provided additional words in the revised manuscript to emphasise this point (see lines 314-318).

**Response to reviewer 2**

*Please note all line references refer to the revised document with tracked changes.*

***This manuscript tested a time varying model parameter framework at a river basin under significant land cover changes in the last few decades. The employed framework is based on Locally Linear Dual EnKF proposed by the authors in the previous studies and is applied to the two conceptual hydrologic models (i.e. HBV and HyMOD).***

***The manuscript shows interesting result and well written in general. However, I have concerns regarding practical applications of the tested approach and the objectives of this research from the following aspects.***

We thank the reviewer for their time and overall positive assessment of the manuscript.  We are also grateful for the constructive comments to make clear the contributions of our work. Please see below responses to specific comments.

***1. In the abstract, the authors stated "rapid land use change impacts on catchment hydrology" and "therefore modeling methodology of such change is important" in the first and second sentences. First of all, please clarify "for what purpose" such modelling representing the land cover change is thought to be needed. This could be, for example, estimating future water resources under further land cover and climate changes or identifying the physical mechanisms of the impact of land cover change on hydrology. Please explain in the introduction for what objective the authors think the modeling with time varying parameter is necessary.***

We appreciate the reviewer's comments that highlight the need to more clearly state the objectives/potential applications of the time varying parameter modelling method.  We have provided additional discussion to ensure this is made clear.  One of the main purposes is to propose a modelling methodology that can provide accurate predictions even when the catchment conditions are changing in real time, as traditional modelling methods are poorly suited to this task (see Lines 33-35 and 55-57). This is useful for forecasting purposes (e.g. flood forecasting) and also for real time reservoir operation in water resource management (see Lines 35-40 and 75-81).  It can also be used to retrospectively assess how model parameters vary in time as a result of land cover changes (see Line 78).

***2. Based on the firs point, please explain how the applied framework with the EnKF can achieve the objectives. Obviously the presented approach requires a full set of input and output to estimate the parameter changes. Suppose that this approach now successfully estimates the time-varying parameters in the HBV model, how can this information be useful for water management, given already land cover change has happened and streamflow change has been already detected in the actual catchment.***

As discussed above, the Locally Linear Dual EnKF can be useful for retrospective analysis of variations to model parameters and also for predictive purposes.  From a prediction perspective, the framework is useful for short term predictions of hydrologic variables in catchments that are undergoing change, where no knowledge of the change is available.  The reason it can be used in this context is because parameters are being updated on-the-fly, at the observation time scale (see Lines 64-66, 73-75). In other words, states and parameters are sequentially changed in real time, whenever an observation becomes available. The most recent observations are then used to make short term predictions of states and parameters (Steps 1-3 in Section 3.2.1). We have provided additional discussion to help clarify how the Locally Linear Dual EnKF can be used for prediction (see Lines 81-82 and 328-332).

***3. Related to the above point, please state clearly the main objective of this research in the introduction. Is the main objective here to test the time varying model parameter framework in the data limited catchment? In such case, what is the criteria to conclude the objective has been achieved. The EnKF may show the parameter changes, but is it enough to validate the method? Or is the main objective here to compare the two model structures?***

The main objectives of this research are two-fold:
1) To investigate the efficacy of the time varying parameter method in a medium sized catchment with realistic land cover change. This goes beyond previous work which focused on small experimental catchments that had drastic land use changes that are easier to infer from streamflow, and are not representative of realistic studies; and
2) To highlight the importance of the chosen model structure in ensuring the success of the time varying parameter method.

We have modified the introduction so that these objectives are explicitly stated (see Lines 89 - 113), along with providing more discussion on the second issue relating to model structure selection (see in particular Lines 109-113).

In regards to validation of the time varying parameter method, its efficacy was assessed based on the ability to represent all aspects of the streamflow hydrograph. In Section 4, we showed that time varying parameter HBV model provided a good representation of baseflow, total and direct runoff (see Figure 7 and 8, and discussion in Lines 445-469). We also showed that the time varying parameter HyMOD model did not represent all aspects of the hydrograph. This showed the importance of the chosen model structure in determining the success of the time varying parameter method.

***With the above main review comments, I have the followings minor comments.***

***1. Abstract L18 and L57: "it serves as an effective tool for separating the influence of climatic and land use change": is this really true? As a result of the EnKF, it is possible that the both effects of land cover and climate changes may be reflected in the wrong way. Given such an ill-identified potential, please explain the logic and actual steps to distinguish the impacts of the two changes.***

We acknowledge that the wording in the abstract is misleading and does not accurately describe the utility of the analysis in Section 4. Our intent is to demonstrate that the time varying parameter approach can be used to show that observed changes to streamflow are not solely a result of changes in meteorological patterns. This is in reference to the analysis undertaken in Section 4 where a resampled rainfall and temperature time series was used with the time varying parameter HBV model (see Lines 471-494). It was found that the resulting simulated streamflow largely reproduced the changes in the observed streamflow (see Lines 486-487 and Figure 8d and h), thereby lending further weight to the conclusion that land cover change has impacted catchment hydrology. This is intended to also demonstrate that the parameter changes correspond to actual changes in catchment hydrology, and are not simply random changes that happen to reproduce the observed streamflow only when the observed forcing time series is used (this discussion has now been inserted into the manuscript, see Lines 492-494).

We have modified the abstract, introduction and conclusion so that the intended message is clearer (see Lines 17-21, 66-70, 504- 507). We have also provided additional discussion explaining the range of plausible conclusions from our analysis, based on the helpful comments from both Reviewer 2 and the Editor (see Lines 479-484). Specifically, it cannot be used to precisely separate out the contributions of forcing and land use change; the analysis can only be used to show that either 1) meteorological changes are the main driver of streamflow changes or 2) land cover changes are contributing to hydrologic change (but potentially in addition to meteorological changes due to the presence of ecosystem feedbacks, as noted by the Editor).

**2. P7 L122 Subsection of "2.1 " may be eliminated because no "2.2" exists.**

An additional sub-heading: "2.1 Data & Land Cover Change" has been inserted to separate the different topics of discussion. The previous Section 2.1 subheading is now Section 2.2.

**3. P10 L206 Is the covariance matrix (sigma) also updated in the sequential Kalman Filter algorithm? Please clarify this part and show the equation if it is also updated.**

The term $\Sigma_{t-1}^{\theta}$ is the sample covariance of the updated (or posterior) parameter ensemble $\{\theta_{t-1}^{i+}\}_{i=1:n}$ at time *t-1*. In Ensemble Kalman Filtering, the covariance matrix is not explicitly updated, but is replaced by the sample covariance of the updated parameter/state members. The equation for updating the members is given in equations 1 and 3 of the original manuscript.

An equation defining $\Sigma_{t-1}^{\theta}$ has been provided (see equation 2 in revised manuscript).

**4. P10 L218 The same comment is also applied to the Kalman gain of the model parameters.**

We assumed this comment is in reference to the sample covariance matrices used to calculate the Kalman gain. Similarly, we have provided equations specifying how these covariance matrices are calculated (see equations 11 and 12 in revised manuscript) and emphasised that it is the sample covariance (see e.g. line 304).

**5. P12 L261 How did you select the tuning parameter s2? The used values in this manuscript should be shown. Table 4 shows "Initial s2 (VVM)" which confuses me because I thought that s2 were set as constant value for each parameter.**

We have expanded the discussion on how the $s^2$ was tuned. In the original submission, we indicated that it was tuned based on the log score of forecast streamflow. We have expanded this discussion to provide more details on this, including equations (see Lines 356-363). The values adopted are shown in Table 4, and are constant values. The column header in Table 4 has been changed to just "$s^2$." We apologise for the error here.

**6. P17 "a MASH undertaken..." Please add a brief explanation of the MASH approach.**

[revised manuscript text omitted]

---

## Referee Report (RR1)

This study applies the time varying parameter method previously developed by the authors to a Vietnamese catchment and two lumped daily hydrological models. The authors test the suitability of their method to reflect observed land use changes within the catchment as well as the compatibility of the method with different model structures. The manuscript is well written, the results very interesting and I appreciate the author's efforts to present their method in a very clear and concise manner. That said, I consider the manuscript can still be improved on several aspects.

Please note that all line numbers refer to the submitted document without changes tracked.

**Key comments**

1) The reader could benefit from more precise explanations on the following points.
The fact that the method is applied to two lumped, conceptual, daily models needs to be stated from the beginning (abstract and introduction) of the article. These are specific methodological choices and could impact the conclusions.
The scope of the paper needs to be more clearly stated by underlining what research gap this study fills (i.e. how your specific contribution will advance understanding) and the novelty of the approach (i.e. what can the time variable parameters method do that existing methods can't when studying the impacts of land use changes).
The perspectives of the study could be better articulated with the paper's scope and better motivated given the outputs of the study. More specifically, the authors propose to apply the time varying parameter method (TVPM) to physically-based models. However, the lines 294-296 state that parameter dimensionality can be an issue and, as acknowledged by the authors, physically-based models are usually less parsimonious than conceptual models. Likewise, the other perspective is to applied the TVPM within a multi-model framework. According to the findings of the analysis, model structure is a key factor in assuring the success of the time varying parameter method: wouldn't it be the same problem to find a single model compatible with the TVPM than to find a compatible multi-model?

2) The temporal scales in the introduction and throughout the manuscript need to be defined more consistently.
Please quantify : L53: "short-term" (one time step ahead/days/week/month?), L54 "dynamic" (daily dynamic/weekly…?), L63 and 71: "real time", L72: "given time", L87: "gradual", L288: "longer time horizons".
The pre-change conditions are different between L206-207 (1973-1979) and Table 1 (1970-1994).
The observed results (Figure 2) are presented between 1970 and 2004 when the modeling results (Figure 3) are presented for the 1975-2004 period.
Likewise why calibrate the models between 1973 and 1979 and not between 1970 and 1994?
It is quite difficult in the present manuscript to gather the different time resolutions.

3) Section 2.2 mixes methods with results.
I would suggest to keep the methodological parts (computation of the base flow index, description of the MASH method and the Mann-Kendall test) as section 2.2 and move the result parts (analysis of

figure 2) as a new section 3.1. It would also be easier for the reader to recall the outputs of the observed changes analysis while moving to the analysis of the time varying parameter method (L367: "as discussed in section 2.2").

Regarding the computation of the BFI please consider adding the equation as well as the chosen values for the two parameters to the text as it can impact the BFI values.

4) The benchmark used in this study appears quite weak for two reasons.

First, the study is retrospective which means both the benchmark and the TVPM should be based on the whole streamflow record. Secondly, the authors mentioned the use of split sample calibration for retrospective studies in the introduction (lines 47-49), why not choose a benchmark based on split sample calibration? The use of such a benchmark could better highlight the benefits of the TVPM over existing methodologies. In particular, it could supplement the discussion the authors provided on the benefits of updating both parameters and states over updating solely the model parameters.

If changing the benchmark is not feasible, the results analysis and discussion should at least acknowledge that better-performing benchmarks already exist and nuance the relative assessment of the efficacy of the TVPM accordingly.

5) I believe the paper could benefit from a more detailed discussion on two aspects.

Could you please expand the explanation of the observed increase of BFI with regard to the physical processes involved. Indeed as stated by the authors, forest coverage decrease for the benefit of cropland. If this is the case, I would expect an observed decrease of BFI since forests usually favor infiltration while cropland are usually characterized by more compact soils and managed to maximize the use of soil water by crops. Are these newly agricultural soils drained or irrigated? It could result respectively in increased soil infiltration and increased available water without changes in the precipitation signal.

Provide some more context to evaluate the results on the model structure impact.

On Figure 3 please ensure that all parameters and states are represented, at least those involved in the TVPM. For example the b parameter (HyMOD) is primarily impacted by the TVPM but not presented in the model scheme so that the reader cannot understand how it is used by the model. Be more specific in the legend of Figure 3: for example, on Fig 3b there is a $q_b$ in the legend but none in the scheme, it is also unclear whether sowat, stw1, Sq1… are the store names or the store content (i.e. the state variable to be updated)? On Figure 5, there is a $k_b$ parameter which is not displayed on Figure 3. If possible, please display parameters using one color and states using another color to help the reader understand model structure quickly.

For the HBV model, perc and β are the two most heavily impacted by TVPM but are also the two most sensitive. I do not find surprising that TVPM would preferably adjust sensitive parameters but a discussion of the relation between model sensitivity and effects of TVPM is missing. To this aim, it would also be very interesting to have the results of the sensitivity analysis for the HyMOD model. Which lead to the following point.

Can the authors elaborate on lines 399-401: "The annual runoff and annual direct runoff are severely under-estimated in the post-change period by the TVP-HyMOD, whilst the Annual Baseflow Index has an increasing trend of magnitude far greater than observed (Figure 7c)."? As stated by the authors (l191-192), the three cascading tanks represent quick flows while slow flow is represented by the $S_s$ store. In Figure 5 the mean alpha parameter is inferior to 0,5 in the post-change period, meaning more flow is routed through the slow flow store, hence the increase of BFI in Figure 7c. My understanding of

these results is that it is easier for the model to adjust its response (simulated streamflow) by modifying the $S_s$ store behavior than to adjust the quick flow response. This could be due to: (i) a high model sensitivity towards ks (especially when alpha is low and b high) and/or (ii) incompatibility between cascading tanks (need of multiple time steps to have an impact on streamflow) and data assimilation frameworks (Markov chain). If this is indeed the case, I would argue that based on their results, the authors should make some concrete recommendations on which type of model structure is compatible with TVPM (parallel tanks, high sensitivity for all parameters, low parameter cross correlation...)

With the above key review comments I have the following minor comments.

**Minor comments**

Line 64-67: "It can also...an assessment." The link with the above paragraph is not obvious at this point of the introduction

Line 72: "given time", do you mean in forecasting mode?

Lines 74-76: please rephrase "the time scale of the observation frequency"

Lines 75-77: Regarding the applications of the method for 1): please clarify the advantages of the approach compared to existing split sample calibration procedures you mentioned (l 48-49), 2) and 3): seam out of the paper scope since the method/results do not include a part on forecasts. Please justify more clearly the use of the method for forecasting. Regarding 3) is on-line water resource water management on the same time scale as the time varying parameter method?

Line 103: Is the efficiency of the method dependent on catchment size? Please specify in the text.

Line 109: Please specify to which dates you are referring

Line 134: Could you explain the reason behind using two different data sets to assess land use? Are the two datasets equally reliable? Please specify in the text

Line 143: Can you describe the variation of altitude within the catchment, as it can help understand the uncertainties associated with the meteorological forcing.

Line 158: Please insert the BFI equation and specify the chosen values for the two parameters

Line 182: Please specify that the daily time step is used

Lines 205-206: Did you use both algorithms on each model or the SCE was used to calibrate HBV and BEA for HyMOD (or reversed)? If a different algorithm was used to calibrate the models, please include the importance of the calibration procedure in the discussion of your results

Line 210: Can you explain why these streamflow threshold values were retained?

Line 247 (eq1): Please name $\mathbf{m_t}$

Line 254: Is $\mathbf{m_{max}}$ the same as the "allowable rate of change" in tables 4 and 5? If yes please unify the notations. Could you also specify how $\mathbf{m_{max}}$ is set (experience with the model, external data…)?

Line 295: Is a large number of parameters a limit to the application of the method? If yes, please acknowledge it in the text

Line 296: Could you briefly explain the Sobol method?

Line 361: Please refer to Figure 4

Lines 375-376: Is the problem the difference between dry and wet seasons or catchment size and heterogeneity? Please clarify.

Lines 379-380: "increased difficulty in accurately modeling the hydrologic response (even in pre-change conditions)": does this mean bad calibration for both models? Please clarify

Line 412: Can the extreme updated values be prevented with smaller allowable change values?

Line 451: "the time varying parameter method"

Line 463: "(i.e. model equation)" could maybe be moved to the beginning of the article to help the reader

Line 464: Is HyMOD unsuited or the association of the time varying method with the HyMOD structure proves inefficient?

Line 466: "unknown future": please rephrase since (i) data assimilation cannot be performed without streamflow measurements ("unknown") and (ii) the "future" has not been explored in this study

References: The formatting of the doi appears different between the citations.

Line 644: Table 1: please add the mean observed BFI values in the Hydro-Meteorological Properties since it is a key variable in your study

---

## Author Response (AR2)

**Response to the Editor**

Dear Dr. Hrachowitz,

Thank you for evaluating our manuscript and review responses and providing constructive comments. We have modified our manuscript to address all review comments. We hope that the modifications to the manuscript help clarify the research gaps and novelty of our work. Specifically, the research gaps are (please note all line references refer to the document with tracked changes):

1) To test the efficacy of the time varying parameter method for realistic catchments that are more heterogeneous, larger, and with more gradual land use change than the test catchments used to demonstrate the proof of concept in *Pathiraja et al.* [2016b]. This is discussed in Lines 93-118 of the revised manuscript.
2) To examine the role of the hydrologic model in determining the success of the time varying parameter approach. This is discussed in Lines 118-124 of the revised manuscript.

The research questions are also summarised in the conclusions (see Lines 647-652).

In regards to the novelty of the study compared to other studies, we have inserted the following discussion (lines 137-139):

*"This work represents the first application of a continuously time varying parameter approach for modelling a real medium sized catchment with no apriori (or partial) knowledge of the type and timing of land use change."*

Additionally, we have discussed the novelty of the approach also in terms of the advantages of the proposed approach over existing methods:

Lines 55-58: *"However, the aforementioned approaches are unsuited to hydrologic forecasting in changing catchments, as the predicted land use change may not reflect actual changes. A potentially more suitable approach in such a setting is to allow model parameters to vary in time, rather than assuming a constant optimal value or stationary probability distribution."*

Lines 82-85: *"In retrospective mode, the method is advantageous compared to split-sample calibration type approaches since no apriori knowledge of land use change is needed, and the modeller does not have to make somewhat arbitrary decisions about how to segregate the data."*

And more specifically, the novelty/advantage of the proposed time varying parameter approach compared to other methods that also utilise the notion of time varying parameters:

Lines 58-64: *"Many existing methods utilising such a framework require some apriori knowledge of the land use change in order to inform variations in model parameters (see for instance Efstratiadis, 2015; Brown et al., 2006; and Westra et al., 2014). Recent efforts have examined the potential for time varying parameter models to automatically adapt to changing conditions using information contained in hydrologic observations and sequential Data Assimilation, without requiring explicit knowledge of the changes [see for example Taver et al., 2015, Pathiraja et al., 2016a&b]."*

In regards to the request for an improved benchmark, we respectfully note that we have not used a benchmark in our study (and a benchmark is not needed for the analyses that we are undertaking).

Full details of our revisions can be found in the response to reviewer document.

Thank you for your time and consideration.

Best Regards,

Sahani Pathiraja
Daniela Anghileri
Paolo Burlando
Ashish Sharma
Lucy Marshall
Hamid Moradkhani

**Response to Reviewer 1**

Please note that all line references refer to the document with tracked changes. Modifications to the manuscript are shown in below in blue.

*This study applies the time varying parameter method previously developed by the authors to a Vietnamese catchment and two lumped daily hydrological models. The authors test the suitability of their method to reflect observed land use changes within the catchment as well as the compatibility of the method with different model structures. The manuscript is well written, the results very interesting and I appreciate the author's efforts to present their method in a very clear and concise manner. That said, I consider the manuscript can still be improved on several aspects.*

We thank the reviewer for their time and comments. Please see below our responses to the comments.

*1) The reader could benefit from more precise explanations on the following points. The fact that the method is applied to two lumped, conceptual, daily models needs to be stated from the beginning (abstract and introduction) of the article. These are specific methodological choices and could impact the conclusions.*

The following text has been inserted in the abstract and introduction:

At line 10: *"The method was used with two lumped daily conceptual models (HBV and HyMOD) that gave good quality streamflow predictions during pre-change conditions."*

At line 131: *"We also consider two lumped conceptual hydrologic models (given the availability of point rainfall, temperature, and streamflow data) operating at daily time step to address the second objective."*

*The scope of the paper needs to be more clearly stated by underlining what research gap this study fills (i.e. how your specific contribution will advance understanding) and the novelty of the approach (i.e. what can the time variable parameters method do that existing methods can't when studying the impacts of land use changes).*

We have modified the introduction of the manuscript so that the research gap is more explicitly defined. The research questions we are examining in this paper are:

1) To test the efficacy of the time varying parameter method for realistic catchments that are more heterogeneous, larger, and with more gradual land use change than the test catchments used to demonstrate the proof of concept in *Pathiraja et al.* [2016b]. This is discussed in Lines 93-118:

*"Here we investigate two issues related to the use of time varying parameter models for prediction in realistic catchments with changing land cover conditions. Firstly, we investigate the efficacy of the time varying parameter method for sparsely observed, medium-sized catchments with spatially complex and gradual land use change (occurring over months/years). Several authors have demonstrated that impacts of land use change on*

*the hydrologic response are dependent on many factors including the type and rate of land cover conversion as well the spatial pattern of different land uses within the catchment [Dwarakish & Ganasri, 2015; Warburton et al., 2012].  In such situations, the effects of unresolved spatial heterogeneities in model inputs (e.g. rainfall) and the relatively less pronounced changes in land surface conditions make time varying parameter detection and accurate hydrologic prediction more difficult."*

2) To examine the role of the hydrologic model in determining the success of the time varying parameter approach.  This is discussed in Lines 118-124:

*"The second objective is to examine the role of the hydrologic model in determining the ability of the time varying parameter framework to provide high quality predictions in changing conditions.  Often there may be several candidate hydrologic models (with time invariant parameters) that have similar predictive performance for a catchment when calibrated and validated over a time series of static land cover conditions.  This work examines whether all such candidate models in time varying parameter mode are also capable of providing accurate predictions under changing conditions."*

The research questions are also summarised in the conclusions (see Lines 647-652).

In regards to the novelty of the study compared to other studies, we have inserted the following discussion (lines 137-139):

*"This work represents the first application of a continuously time varying parameter approach for modelling a real medium sized catchment with no apriori (or partial) knowledge of the type and timing of land use change."*

Additionally, we have discussed the novelty of the approach also in terms of the advantages of the proposed approach over existing methods:

Lines 55-58: *"However, the aforementioned approaches are unsuited to hydrologic forecasting in changing catchments, as the predicted land use change may not reflect actual changes.  A potentially more suitable approach in such a setting is to allow model parameters to vary in time, rather than assuming a constant optimal value or stationary probability distribution."*

Lines 82-85: *"In retrospective mode, the method is advantageous compared to split-sample calibration type approaches since no apriori knowledge of land use change is needed, and the modeller does not have to make somewhat arbitrary decisions about how to segregate the data."*

And more specifically, the novelty/advantage of the proposed time varying parameter approach compared to other methods that also utilise the notion of time varying parameters:

Lines 58-64: *"Many existing methods utilising such a framework require some apriori knowledge of the land use change in order to inform variations in model parameters (see for instance Efstratiadis, 2015; Brown et al., 2006; and Westra et al., 2014).  Recent efforts have examined the potential for time varying parameter models to automatically adapt to*

*changing conditions using information contained in hydrologic observations and sequential Data Assimilation, without requiring explicit knowledge of the changes [see for example Taver et al., 2015, Pathiraja et al., 2016a&b]."*

**The perspectives of the study could be better articulated with the paper's scope and better motivated given the outputs of the study. More specifically, the authors propose to apply the time varying parameter method (TVPM) to physically-based models. However, the lines 294-296 state that parameter dimensionality can be an issue and, as acknowledged by the authors, physically-based models are usually less parsimonious than conceptual models. Likewise, the other perspective is to applied the TVPM within a multi-model framework. According to the findings of the analysis, model structure is a key factor in assuring the success of the time varying parameter method: wouldn't it be the same problem to find a single model compatible with the TVPM than to find a compatible multi-model?**

The discussion surrounding physically based models and multi-model framework in the conclusion was aimed at providing *potential* solutions to the issue of model specification. We have provided additional discussion regarding physically based models. Specifically, that the dimension of the time varying parameter vector may need to be reduced to make the estimation problem tractable, and that models of intermediate complexity may be more promising (see lines 671-681):

*"One possible way to ensure success of the time varying parameter approach is to use models whose fundamental equations explicitly represent key physical processes (for instance, modelling sub-surface flow using Richard's equation with hydraulic conductivity allowed to vary with time). In this way, time variations in model parameters would more closely reflect changes to physiographic properties, rather than also having to account for missing processes. The drawback of such physically based models is that they are generally data intensive, both in generating model simulations (i.e. detailed inputs) and specifying parameters. Additionally, it may be necessary to reduce the dimensionality of the time varying parameter vector by keeping less sensitive model parameters fixed in order to make the estimation problem tractable. Models of intermediate complexity that have explicit process descriptions may be the most promising, although this also remains to be demonstrated."*

The discussion regarding a multi-model framework has been removed. The idea here was that a suite of models would be used (e.g. in this case both HBV and HyMOD, since both gave reasonable simulation performance in pre-change conditions) and any model that was unable to represent key features of the hydrologic response would be given less weight (in this case, HyMOD).

**2) The temporal scales in the introduction and throughout the manuscript need to be defined more consistently. Please quantify : L53: "short-term" (one time step ahead/days/week/month?), L54 "dynamic" (daily dynamic/weekly...?), L63 and 71: "real time", L72: "given time", L87: "gradual", L288: "longer time horizons".**

Short-term: days to weeks, this has been added: *"2) for short-term predictive modelling (days to weeks), e.g. flood forecasting;" (line 80)*

Dynamic: this word was used to refer to catchments whose properties are changing with time.  This has been replaced with "changing." (line 55)

Real time: this is a commonly used term to refer to "at the actual time the process is occurring."

Given time: this was meant to refer to "at each time in the assimilation cycle."  This phrase has been deleted. (line 73)

Gradual:  The following text has been inserted: *"medium-sized catchments with spatially complex and gradual (occurring over months/years) land use change." (lines 95-96).*

Longer time horizon: in this context, this phrase is referring to forecasts at longer than one time step ahead.  The following text has been inserted: *"Forecasts at longer time horizons (i.e. longer than one time step ahead) would be made by generating prior parameters and states as detailed in Steps 1 to 3,…"* (line 401).

***The pre-change conditions are different between L206-207 (1973-1979) and Table 1 (1970-1994). The observed results (Figure 2) are presented between 1970 and 2004 when the modeling results (Figure 3) are presented for the 1975-2004 period.  Likewise why calibrate the models between 1973 and 1979 and not between 1970 and 1994?***

***It is quite difficult in the present manuscript to gather the different time resolutions.***

We apologise for the confusion in this regard and have made the following clarifications.

The data in Table 1 is presented for pre and post 1994 based on the available land cover map information.  Hence 1970-1994 is taken as the entire pre-change period and post 1994 as the post-change period.  The following text has been inserted to clarify (see Lines 162-164): *"A summary of catchment properties is provided in Table 1 for pre-change (prior to 1994) and post-change (after 1994) conditions.  This separation was based on available land cover information as described below."*

Only part of the pre-change period was selected for calibration, since it is of interest to undertake assimilation on pre-change data also (to see if parameters stay constant).  We have modified the text to make clear that the period 1973-1979 is only a part of the pre-change period (see lines 292-294): *"The period 1973 to 1979 was selected for calibration (with 2 years for spin-up) as it was expected to have minimal land cover changes (and is therefore representative of pre-change conditions), and also to ensure sufficient data on pre-change conditions is available for assimilation."*

The observed data have been analysed for the entire period of record in Figure 2, since here we are interested in presenting statistics for the entire data set.  This is needed so we can determine when changes occur, as discussed in Lines 179-181: *"Based on the available land cover map information and the changes to observed runoff (see Section 2.2), we posit that a period of rapid extensive deforestation occurred in early to mid-1990s."*

Finally, Figure 4 and 5 contain results of the assimilation for the period after calibration (1980 to 2004). These have been modified so that they show the results from 1980 to 2004.

As mentioned earlier, it is of interest to undertake time varying parameter estimation even in the pre-change conditions (up to 1994) to see if it is able to detect constant parameters during the period of minimal change. Significant parameter variations in during this period indicate the presence of model structural issues.

*3) Section 2.2 mixes methods with results. I would suggest to keep the methodological parts (computation of the base flow index, description of the MASH method and the Mann-Kendall test) as section 2.2 and move the result parts (analysis of figure 2) as a new section 3.1. It would also be easier for the reader to recall the outputs of the observed changes analysis while moving to the analysis of the time varying parameter method (L367: "as discussed in section 2.2"). Regarding the computation of the BFI please consider adding the equation as well as the chosen values for the two parameters to the text as it can impact the BFI values.*

We thank the reviewer for the suggestion regarding Section 2.2, but feel that its present state is most appropriate since the aim here is to provide a discussion on the impact of land cover change, prior to undertaking the time varying parameter estimation which is the main focus of this manuscript.

The recursive filter used to estimate baseflows has been inserted (see equation 1), as well as the values of the 2 parameters (see Line 198). The equation for the annual baseflow index was provided in Line 218.

*4) The benchmark used in this study appears quite weak for two reasons. First, the study is retrospective which means both the benchmark and the TVPM should be based on the whole streamflow record. Secondly, the authors mentioned the use of split sample calibration for retrospective studies in the introduction (lines 47-49), why not choose a benchmark based on split sample calibration? The use of such a benchmark could better highlight the benefits of the TVPM over existing methodologies. In particular, it could supplement the discussion the authors provided on the benefits of updating both parameters and states over updating solely the model parameters. If changing the benchmark is not feasible, the results analysis and discussion should at least acknowledge that better-performing benchmarks already exist and nuance the relative assessment of the efficacy of the TVPM accordingly.*

We are unclear as to what exactly the reviewer is referring to when they discuss "the benchmark." In this manuscript, we are analysing the output from the time varying parameter estimation algorithm only, we have made no reference to a benchmark. Additionally, we are unclear about the reviewer's request to undertake TVPM on the whole streamflow record. We have undertaken the time varying parameter estimation on the period 1979 to 2004, which is almost the entire streamflow record.

In regards to the reviewer's request to examine split sample calibration: we respectfully note that the purpose of this article is to examine specific application issues related to the use of the TVPM, not to highlight its benefit over existing methodologies. The scope of the article is discussed in Lines 93-124, which is:

1) To test the efficacy of the time varying parameter method for realistic catchments that are more heterogeneous, larger, and with more gradual land use change than the test catchments used to demonstrate the proof of concept in *Pathiraja et al.* [2016b].

2) To examine the role of the hydrologic model in determining the success of the time varying parameter approach.

Secondly, split sample calibration is not a suitable benchmark here because we are focused on modelling approaches that can also be used in forecasting and predictive mode, without any *apriori* knowledge of the catchment changes as stated in Lines 82-85:

*"In retrospective mode, the method is advantageous compared to split-sample calibration type approaches since no apriori knowledge of land use change is needed, and the modeller does not have to make somewhat arbitrary decisions about how to segregate the data."*

**5) I believe the paper could benefit from a more detailed discussion on two aspects.**

**Could you please expand the explanation of the observed increase of BFI with regard to the physical processes involved. Indeed as stated by the authors, forest coverage decrease for the benefit of cropland. If this is the case, I would expect an observed decrease of BFI since forests usually favor infiltration while cropland are usually characterized by more compact soils and managed to maximize the use of soil water by crops. Are these newly agricultural soils drained or irrigated? It could result respectively in increased soil infiltration and increased available water without changes in the precipitation signal.**

Unfortunately, not much information about the agricultural practices in the region is available, but, to our knowledge, there are no significant water storing facilities in that region which could support extensive irrigation schemes. We included the following discussion about the physical processes potentially involved with BFI increase (lines 222-230):

*"The exact physical processes behind the observed increase in baseflow are not precisely known, particularly since effects of land use change from forest to cropland are not unequivocal [Price, 2011]. Deforestation may be associated to an increase in mean annual flow and baseflow because of lower interception and evapotranspiration rates [e.g., Keppeler and Ziemer, 1990]. Nevertheless, permanent forest removal may decrease baseflow because of soil compaction and lower infiltration rates [e.g., Zimmermann et al., 2006; Bormann and Klaassen; 2008]. Some authors also show that tillage practices, associated to forest conversion to cropland, can increase soil porosity, soil water content, and infiltration, thus ultimately contributing to baseflow formation [e.g., Alam et al., 2014]."*

**Provide some more context to evaluate the results on the model structure impact. On Figure 3 please ensure that all parameters and states are represented, at least those involved in the TVPM. For example the b parameter (HyMOD) is primarily impacted by the TVPM but not presented in the model scheme so that the reader cannot understand how it is used by the model. Be more specific in the legend of Figure 3: for example, on Fig 3b there is a qb in the legend but none in the scheme, it is also unclear whether sowat, stw1, Sq1... are the store names or the store content (i.e. the state variable to be updated)? On**

*Figure 5, there is a kb parameter which is not displayed on Figure 3. If possible, please display parameters using one color and states using another color to help the reader understand model structure quickly.*

Thank you for the suggestions to improve Figure 3. All states and parameters have now been included in Figure 3 and the naming of the parameters (e.g. kb vs ks) has now been made consistent both within the text and between Figure 3 and Figure 5. States and parameters have also been represented in different colours in Figure 3 to make each clearer.

*For the HBV model, perc and β are the two most heavily impacted by TVPM but are also the two most sensitive. I do not find surprising that TVPM would preferably adjust sensitive parameters but a discussion of the relation between model sensitivity and effects of TVPM is missing.*

A discussion on sensitivity and correlation with the observed variables has been provided (see Lines 522-526):

*"These changes correspond with the observed increase in the annual runoff coefficient (Figure 2) and increase in baseflow volume (as discussed in Section 2.2). From an algorithm perspective, these parameters are most strongly correlated with streamflow (as well as the most sensitive, see Table 3), meaning that they will receive the greatest proportional updates."*

*To this aim, it would also be very interesting to have the results of the sensitivity analysis for the HyMOD model. Which lead to the following point. Can the authors elaborate on lines 399-401: "The annual runoff and annual direct runoff are severely under-estimated in the post-change period by the TVP-HyMOD, whilst the Annual Baseflow Index has an increasing trend of magnitude far greater than observed (Figure 7c)."? As stated by the authors (l191-192), the three cascading tanks represent quick flows while slow flow is represented by the Ss store. In Figure 5 the mean alpha parameter is inferior to 0,5 in the post-change period, meaning more flow is routed through the slow flow store, hence the increase of BFI in Figure 7c. My understanding of these results is that it is easier for the model to adjust its response (simulated streamflow) by modifying the Ss store behavior than to adjust the quick flow response. This could be due to: (i) a high model sensitivity towards ks (especially when alpha is low and b high) and/or (ii) incompatibility between cascading tanks (need of multiple time steps to have an impact on streamflow) and data assimilation frameworks (Markov chain). If this is indeed the case, I would argue that based on their results, the authors should make some concrete recommendations on which type of model structure is compatible with TVPM (parallel tanks, high sensitivity for all parameters, low parameter cross correlation...)*

We have provided additional discussion to clarify the interpretation of the estimated time varying parameters in the HyMOD. The reviewer is correct in identifying that the alpha parameter is reduced below 0.5 in the post-change period, so that more water is routed through the slow flow store. However, the reason for this is due to the observed increase in persistent flows during periods of no rain, and the fact that the slow flow is the only active store during such periods, because the quick flow store has been depleted. This means that the only parameters that have any impact on streamflow are $k_s$ and $\alpha$, which is why these are adjusted.  The following discussion has been provided to explain this further (see lines 577-589):

*"The reason for the differences in performance between the TVP-HBV and TVP-HyMOD lies in the structure of the hydrologic model.  The TVP-HyMOD is incapable of representing the observed increase in annual runoff/direct runoff coefficient due to the increased baseflow during dry periods, despite having an Annual Baseflow Index far greater than the observed.  This occurs due to an inability to generate flow volume during periods of no rain.   In joint state-parameter updating using HyMOD, underestimated runoff predictions during dry periods lead to adjustments to the $k_s$ and $\alpha$ parameters to increase baseflow depth (since these are the only parameters that are associated to an active store).  Unlike HBV, HyMOD has no continuous supply of water to the routing stores (i.e. the quick flow and slow flow stores) during recession periods (which typically have extended periods of no rainfall, so that V in Figure 3 is zero).  This means that $k_s$ and $\alpha$ are updated to extreme values to compensate for the volumetric shortfall.  The HBV structure, on the other hand, has a continuous percolation of water into the deep layer store even during periods of no rain (so long as the shallow water store is non-empty)."*

In regards to the reviewer's request to provide concrete recommendations, this is non-trivial because the issue is not the compatibility of the hydrologic model with the TVPM, but rather the suitability of the model to simulate changed streamflow dynamics.  The model structure is incapable of generating persistent flows during periods of no rain, regardless of the parameter setting (as explained above).  The recommendations that we can provide are that a sufficiently flexible model structure must be chosen prior to undertaking TVP in real time.  The following discussion has been inserted (see lines 589-594):

*"In summary, the HyMOD model structure is poorly suited to simulating streamflow dynamics in post-change conditions, although it gave reasonable simulations in pre-change conditions.  This highlights that need to select a sufficiently flexible model structure prior to undertaking forecasting/predictive modelling using the time varying parameter approach.  In particular, the model structure must be capable of effectively simulating all potential future catchment conditions."*

**Minor comments**

*Line 64-67: "It can also...an assessment." The link with the above paragraph is not obvious at this point of the introduction.*

This statement is just adding to the discussion on the capabilities of the method.

*Line 72: "given time", do you mean in forecasting mode?*

Yes, this would be in forecasting mode.

*Lines 74-76: please rephrase "the time scale of the observation frequency"*

This has been replaced with *"at the time scale of the available observations."*

***Lines 75-77: Regarding the applications of the method for 1): please clarify the advantages of the approach compared to existing split sample calibration procedures you mentioned (l 48-49), 2) and 3): seam out of the paper scope since the method/results do not include a part on forecasts. Please justify more clearly the use of the method for forecasting. Regarding 3) is on-line water resource water management on the same time scale as the time varying parameter method?***

Additional discussion on the advantages of the method over split sample calibration has been included (see Lines 82-85):

*"In retrospective mode, the method is advantageous compared to split-sample calibration type approaches since no apriori knowledge of land use change is needed, and the modeller does not have to make somewhat arbitrary decisions about how to segregate the data."*

This is in addition to the discussion in Lines 58-64:

*"Many existing methods utilising such a framework require some apriori knowledge of the land use change in order to inform variations in model parameters (see for instance Efstratiadis, 2015; Brown et al., 2006; and Westra et al., 2014).  Recent efforts have examined the potential for time varying models to automatically adapt to changing conditions using information contained in hydrologic observations and sequential Data Assimilation, without requiring explicit knowledge of the changes [see for example Taver et al., 2015, Pathiraja et al., 2016a&b]."*

Additional discussion on the use of the method for prediction/forecasting has been provided (see Lines 85-89):

*"When used for prediction or forecasting, states and parameters are updated sequentially using all available observations up until the current time.  These updated states and parameters are then used along with the prior parameter generating model to produce hydrologic predictions over a short time horizon. This allows one to seamlessly obtain predictions without the modeller needing to explicitly modify the model to account for any catchment changes."*

The advantage of using the method in forecasting mode compared to existing approaches has also been discussed (lines 53-64):

*"A related approach involves combining land use change forecast models with hydrologic models [e.g. Wijesekara et al., 2012].  However, the aforementioned approaches are unsuited to hydrologic forecasting in changing catchments, as the predicted land use change may not reflect actual changes.  A potentially more suitable approach in such a setting is to allow model parameters to vary in time, rather than assuming a constant optimal value or stationary probability distribution... Recent efforts have examined the potential for time varying parameter models to automatically adapt to changing conditions using information contained in hydrologic observations and sequential Data Assimilation, without requiring explicit knowledge of the changes [see for example Taver et al., 2015, Pathiraja et al., 2016a&b]."*

Finally, when used for on-line water management, this would indeed be at the same time scale as the parameters are updated.  This is reflected by the use of the phrase "real-time" in Line 80.

**Line 103: Is the efficiency of the method dependent on catchment size? Please specify in the text.**

The reference to size here is related to efficacy rather than efficiency, since larger catchments are usually more difficult to model well compared to smaller catchments (particularly with lumped conceptual models).

**Line 109: Please specify to which dates you are referring**

The following has been inserted (see Line 134-135): *"during the pre-change calibration period (1975-1979)."*

**Line 134: Could you explain the reason behind using two different data sets to assess land use? Are the two datasets equally reliable? Please specify in the text**

It was not easy to find (continuous in time and from the same source) land cover maps for that area. These were the only two sources we could find.  The following text has been inserted (see Lines 166-168): *"Land cover information for the catchment is scant, we were able to locate only two sources which unfortunately do not give a complete picture over the entire time period of interest (1970 to 2004)."*

**Line 143: Can you describe the variation of altitude within the catchment, as it can help understand the uncertainties associated with the meteorological forcing.**

The following text has been inserted (see lines 161-162): *"and catchment elevation ranges between 350 and 1500 m asl."*

**Line 158: Please insert the BFI equation and specify the chosen values for the two parameters**

The recursive filter used to estimate baseflows has been inserted (see equation 1), as well as the values of the 2 parameters (see Line 198).  The equation for the annual baseflow index was provided in Line 218.

**Line 182: Please specify that the daily time step is used**

The following text has been inserted (line 258): *"Conceptual lumped models operating at a daily time step…"*

**Lines 205-206: Did you use both algorithms on each model or the SCE was used to calibrate HBV and BEA for HyMOD (or reversed)? If a different algorithm was used to calibrate the models, please include the importance of the calibration procedure in the discussion of your results**

The following discussion has been inserted to clarify how the models were calibrated, and also to note that the calibration procedure is not critical in our study (Lines 286-291):

*"The Shuffled Complex Evolution Algorithm (SCE-UA) [Duan et al., 1993] was used to calibrate HyMOD and the Borg Evolutionary Algorithm [Hadka & Reed, 2013] was used to calibrate HBV. The calibration algorithms were selected based on previous studies that had successfully used them for calibration of these models [Reed et al., 2013; Moradkhani et al., 2005]. The calibration procedure itself is however not critical in our study, because the optimal parameter values are only used as initial values for the time varying parameter method."*

**Line 210: Can you explain why these streamflow threshold values were retained?**

Explanation of how the streamflow threshold values were obtained have been added to the manuscript (see Lines 297-299):

*"Here the low flow threshold was defined as the average annual 50th percentile flow and the high flow threshold as the average annual 85th percentile flow."*

**Line 247 (eq1): Please name $m_t$**

The following text has been inserted (line 352): *"$m_t$, the estimated rate of change."*

**Line 254: Is mmax the same as the "allowable rate of change" in tables 4 and 5? If yes please unify the notations. Could you also specify how mmax is set (experience with the model, external data...)?**

The notation in tables 4 and 5 has been updated to say $m_{max}$. Specifying the max allowable rate of change requires knowledge of the model and some educated judgement as to the likely changes of the catchment. The following text has been inserted (see lines 354-361): *"The maximum rate of change is model specific and will depend on the modeller's judgement regarding expected extreme changes."*

**Line 295: Is a large number of parameters a limit to the application of the method? If yes, please acknowledge it in the text**

The issue of estimating a high dimensional parameter vector from low dimensional data is problematic for any parameter estimation method. The following text has been inserted (lines 419-421): *"Estimating a large number of parameters from limited data is problematic in that the system is highly under-determined, making it difficult to ensure the estimated parameters are meaningful."*

**Line 296: Could you briefly explain the Sobol method?**

We have provided additional discussion on the Sobol method, although the discussion is kept brief since it is a minor step in our study (lines 421-431):

*"Given the fairly low parameter dimensionality of HyMOD, all model parameters were allowed to vary in time whilst for HBV we applied the Sobol method to identify the most sensitive parameters to be included in the time varying parameter estimation. The Sobol*

*method is a global sensitivity analysis method based on variance decomposition. It identifies the partial variance contribution of each parameter to the total variance of the hydrological model output [see for example Saltelli et al., 2008, Nossent et al. 2011]. The method, implemented through the SAFE toolbox [Pianosi et al., 2015], found the $lp$ and $Maxbas$ parameters to be the least sensitive and least important in defining variations to catchment hydrology (see Table 3). These were held fixed ($lp$ = 1 and $Maxbas$ = 1 day) in the following analysis. Note that although the $hl1$ parameter was found to have low sensitivity, it was retained as a time varying parameter due to its conceptual importance in separating interflow and near surface flow (refer Figure 3)."*

**Line 361: Please refer to Figure 4**

The following text has been inserted (line 518): "(see Figure 4 and 5)."

**Lines 375-376: Is the problem the difference between dry and wet seasons or catchment size and heterogeneity? Please clarify.**

The issue is the difficulty in modelling wet and dry season flows, reference to catchment size and heterogeneity has been deleted.

**Lines 379-380: "increased difficulty in accurately modeling the hydrologic response (even in pre- change conditions)": does this mean bad calibration for both models? Please clarify**

This statement is referring to the fact that the streamflow from this catchment is comparatively more difficult to model accurately using the lumped models compared to the smaller catchments referenced in the previous sentence. This is not necessarily just calibration, since there is a portion of the pre-change period that is also considered in the assimilation period.

**Line 412: Can the extreme updated values be prevented with smaller allowable change values?**

The max allowable change value is for proposing prior parameters, whilst this statement is referring to updated parameters. Updated parameters means parameters that are modified by the Kalman update equation (equation 9). Extreme updated values may occur when the prior parameters produce streamflow values that are a poor fit to the observations, thereby requiring large changes to the parameters to which the streamflow is most correlated.

**Line 451: "the time varying parameter method"**

Corrected to (line 649): *"time varying parameter estimation method."*

**Line 463: "(i.e. model equation)" could maybe be moved to the beginning of the article to help the reader**

The statement (i.e. model equations) has been added at line 135-137 in the Introduction: *"Therefore, the effect of the model structure (i.e. model equations) on hydrologic predictions from the time varying parameter models is studied."*

**Line 464: Is HyMOD unsuited or the association of the time varying method with the HyMOD structure proves inefficient?**

The structure of the HyMOD model equations is not suited, as discussed in Lines 577 to 594. The issue relates specifically to the persistent flows during dry flows that occurs only after land use change. The structure of HyMOD is such that there is no continuous supply of water to the routing stores during dry periods or recession periods (Line 584). This means that the $V$ variable in Figure 3 is zero, so that $k_s$ and $\alpha$ have to be set to extreme values in order to generate any outflow (in this time period, the value of the other parameters is irrelevant) (Lines 582-584, 586-587). This issue is entirely a consequence of the model, and would be present even in standard calibration. The structure of HBV is more amenable to producing persistent flows during dry flows, hence this issue is not seen.

**Line 466: "unknown future": please rephrase since (i) data assimilation cannot be performed without streamflow measurements ("unknown") and (ii) the "future" has not been explored in this study**

We respectfully note that this statement is referring to the choice of the model structure, which has to be made before the time varying parameter estimation is carried out. Whilst in this study we have undertaken a retrospective analysis, the same approach can be undertaken in real time, meaning that a model has to be selected before any potential land use change occurs (hence unknown future land use change). We have added the following to clarify this (Line 664-671):

*"This work shows that the chosen model is critical for ensuring the time varying parameter framework successfully models streamflow in unknown future land cover conditions, particularly when used in a real time forecasting mode. Appropriate model selection can be a difficult task due to the significant uncertainty associated with future land use change, and can be even more problematic when multiple models have similar performance in pre-change conditions (as was the case in this study)."*

**References: The formatting of the doi appears different between the citations.**

The formatting of the doi has been made consistent.

**Line 644: Table 1: please add the mean observed BFI values in the Hydro-Meteorological Properties since it is a key variable in your study**

The estimated mean annual BFI has been added to Table 1.

[revised manuscript text omitted]

---

## Author Response (AR3)

Dear Dr. Hrachowitz,

Thank you for the prompt evaluation of our revised manuscript and the positive assessment. In regards to your response regarding the benchmark: yes, we do indeed briefly compare to the time invariant parameter case and this can be considered a benchmark. We apologise for the confusion here; our intent was that no benchmark was used in investigating the impact of the model structure on the performance of the time varying parameter model, which is the main focus of the paper. The majority of the discussion is devoted to comparing the output from state and parameter updating (so called LL Dual EnKF with HBV or HyMOD) and time varying parameter model with no state updating (TVP-HBV and TVP-HyMOD).

We have added in the following statements as per your requests. Specifically, additional discussion has been provided to make clear that the time invariant parameter models were calibrated on pre-1979, and that the comparisons have been made for the time invariant parameter model evaluated over the post 1979 period. Additionally, the improvement has been quantified using the NSE.

In the methods section (Section 3.2.2, lines 348-351 of manuscript with tracked changes):

*"Joint state and parameter estimation was undertaken for the Nammuc Catchment over the period 1980 to 2004 by assimilating streamflow observations into the HyMOD and HBV models at a daily time step. Additionally, simulations using the time invariant parameters obtained from calibration over the period 1973-1979 were generated for 1980 to 2004, for comparison."*

In the results section (Section 4, lines 463-470 of manuscript with tracked changes):

*"Streamflow predictions from the LL Dual EnKF (i.e. with state and parameter updating) for both the HyMOD and HBV are generally of similar quality and superior to those from the respective time invariant parameter models that have been calibrated on pre-change data (1975-1979), although a slight bias in baseflow predictions from HyMOD is evident (see for example Figure 6). The Nash Sutcliffe Efficiency of one step ahead streamflow predictions over the period 1980 – 2004 from the LL Dual EnKF is 0.87 when using HyMOD or HBV, compared to 0.76 and 0.72 for the respective time invariant parameter models evaluated over the same period. However, differences in predictions from TVP-HBV and TVP-HyMOD are more striking due to the lack of state updating."*

We hope this clarifies any ambiguities, and we look forward to your response. Thank you for your time and consideration.

Best Regards,

[revised manuscript text omitted]